# INTERDANCE: REACTIVE 3D DANCE GENERATION WITH REALISTIC DUET INTERACTIONS

## ABSTRACT

Humans perform a variety of interactive motions, among which duet dance is one of the most challenging interactions. However, in terms of human motion generative models, existing works are still unable to generate high-quality interactive motions, especially in the field of duet dance. On the one hand, it is due to the lack of large-scale high-quality datasets. On the other hand, it arises from the incomplete representation of interactive motion and the lack of fine-grained optimization of interactions. To address these challenges, we propose, **InterDance**, a large-scale duet dance dataset that significantly enhances motion quality, data scale, and the variety of dance genres. Built upon this dataset, we propose a new motion representation that can accurately and comprehensively describe interactive motion. We further introduce a diffusion-based framework with an interaction refinement guidance strategy to optimize the realism of interactions progressively. Extensive experiments demonstrate the effectiveness of our dataset and algorithm. Our project page is https://inter-dance.github.io/.

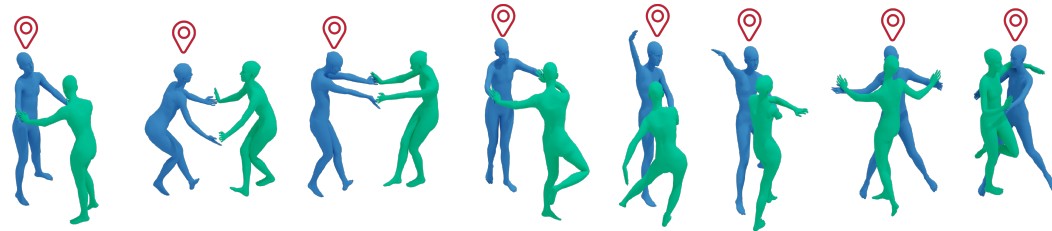

Figure 1: Example of reactive 3D dance generation. Green represents the leader and blue represents the follower (also positioned with a red marker). Given the music and leader's dance, the goal of reactive dance generation is to generate the follower's dance that coordinates with the music and leader.

## 1 INTRODUCTION

Dance is one of the most important elements in film, animation, and game production. The rapid generation of high-quality 3D dances using computer algorithms has significant practical value. In duet dance, two dancers perform interactive dances with music and can be typically divided into a leader and a follower. The leader is responsible for initiating and guiding the dance, while the follower responds to the leader's cues and follows the leader (Siyao et al., 2024). Following this paradigm, we focus on the task of reactive dance generation. Given the music and the leader's dance as inputs, our goal is to generate the follower's dance with accurate interactive movements and physical realism.

However, this task presents significant challenges. Firstly, large-scale, high-quality datasets for duet dances are scarce. Secondly, current algorithms fail to adequately model strong interactions, resulting in generated follower dance movements that lack both interactivity and physical realism. Although some datasets are currently available for solo dances (Li et al., 2023b; 2021a), high-quality duet dance data remains extremely scarce. Le et al. (Wang et al., 2022) introduce the

AIOZ-GDANCE dataset, which includes YouTube videos of 2-7 dancers. They used monocular SMPL pose estimation algorithms to obtain SMPL-format motion. However, the predicted motion is low quality and lacks finger movements because estimating 3D multi-human motion from monocular video is a challenging problem, plagued by issues such as video motion blur and multi-person occlusion. Duolando (Siyao et al., 2024) propose a duet dance dataset DD100 that includes finger movements, but it only contains 1.92 hours of data with 10 ballroom dance genres, making it difficult to train robust and generalizable networks. To further improve the scale and quality of duet dance data, we introduce the InterDance dataset. We employ experienced dancers and professional MoCap equipment to record 3.93 hours of music-paired duet dance. The InterDance dataset includes not only ballroom dances but also folk, classical, and street dances, making a total of 4 major categories and 15 diverse fine-grained genres. InterDance features precise body and finger movements, realistic physical interactions between dancers, and accurate foot-ground contact.

Generating the follower's dance based on the music and the leader's dance has significant challenges in ensuring accurate interactivity and physical realism. Duet dances often involve strong interactions, such as hand-holding and shoulder touches. However, existing methods fail to model these strong interactions accurately. This failure arises partly because current algorithms use internal joint positions to represent motion, inherently lacking surface body information, which results in inaccurate contacts. Additionally, current methods generate interactive movements based solely on learned data priors without incorporating specific feedback on contacts and penetration during the generation process. To address these issues, we propose a new motion representation suitable for reactive dance generation. This representation expresses body and hand movements in canonical space and includes downsampled body surface vertices, along with contact labels to enhance the dancers' interaction accuracy. Moreover, we introduce a diffusion-based reactive dance generation algorithm. This algorithm employs a contact and penetration guided sampling strategy based on contact labels and signed distance fields to improve the accuracy and physical realism of interactions.

Our main contributions are as follows: **(1)** We introduce a duet dance dataset featuring body and finger movements with high physical realism, comprising 3.93 hours of music-paired duet dance across 15 different dance genres. **(2)** We propose a new motion representation. It includes body surface information and contact labels, helping algorithms generate more realistic interactions. **(3)** We propose a diffusion-based reactive dance motion generation model with Interactive Refine Guidance to enhance the accuracy and realism of interactive dance movements.

Code, model, and data will be publicly available.

## 2 RELATED WORKS

**Music-Dance Datasets.** Existing works mainly focus on single-person dance datasets, where movements are either estimated from dance videos using algorithms or captured by professional MoCap equipment and dancers. The former method does not require specialized motion capture equipment, making it easier to implement. Li et al. (Li et al., 2021b) used multiple cameras to capture dancers from different angles, obtaining multi-view dance videos. They then used algorithms to estimate the dance movements, creating the AIST++ dataset of 5.2 hours for solo dance. However, accurately estimating complex dance movements from videos is challenging due to video occlusion and blur. This often results in failures to capture fine-grained finger movements and complex rapid actions such as "Backflip", "Thomas rotation", etc. Professional MoCap Equipment can precisely capture complex movements and is widely used in animation and film production. Therefore, FineDance (Li et al., 2023b) collects a dataset of solo dance performances totaling 14.6 hours, including accurate body and finger movements. Recently, Duolando (Siyao et al., 2024) introduced the DD100 dataset, which comprises only 1.92 hours of duet dance data with paired music. Additionally, ReMoCap (Ghosh et al., 2024), InterHuman Liang et al. (2024), and Inter-X (Xu et al., 2024a) are also high-quality, large-scale Human-Human Interaction datasets. While they include a small portion of the dance, they are more focused on text-driven generation and lack music.

**Dance generation.** Recent advancements in the music-to-dance generation have seen significant progress (Tang et al., 2018; Li et al., 2020; Sun et al., 2020; Zhuang et al., 2022; Sun et al., 2022; Qi et al., 2023). FACT (Li et al., 2021b) utilizes cross-modal transformer blocks with strong sequence modeling capabilities to generate single-person dance from given music. Bailando (Siyao et al., 2022; 2023) introduces a two-stage framework, with the first stage encoding and quantizing

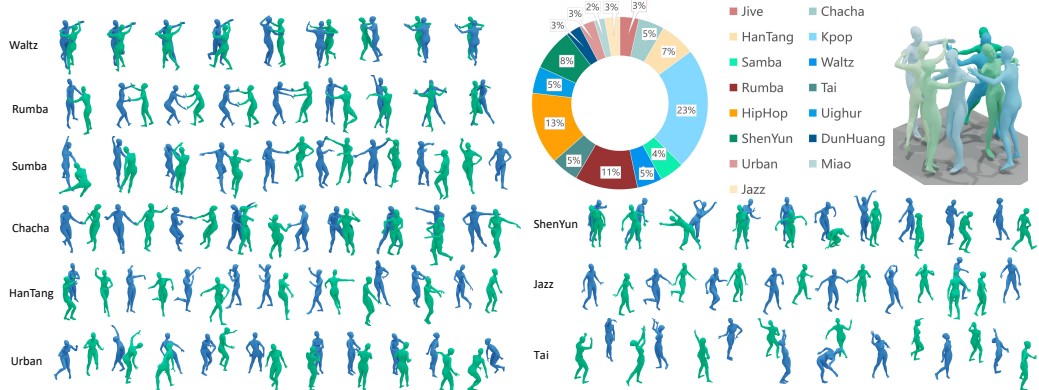

Figure 2: Visualizations of the InterDance samples, the dataset contains high-quality duet dance with accurate body and fingers, there are a total of 15 fine-grained diverse genres.

3D dance and the second stage using an actor-critic GPT to generate dance. EDGE (Tseng et al., 2023) introduces a diffusion-based approach for dance generation with editing capabilities and can generate arbitrarily long sequences. Lodge(Li et al., 2024) designs a two-stage parallel Diffusion architecture for efficient ultra-long dance generation, which can can generate minutes of dance in a few seconds. GCD (Le et al., 2023a) introduces a diffusion-based neural network for music-driven group dance generation with a group contrastive strategy to enhance the connection between dancers and their group. However, existing methods mainly focus on generating solo dances or group dances with only weak interactions. There remains a significant challenge in generating more complex interactive dances.

## 3 THE INTERDANCE DATASET

There is still a lack of **large-scale** duet datasets with accurate **strong interactions**. To address the shortcomings of existing datasets, we collect as much duet dance data as possible and use professional MoCap equipment to accurately record two dancers' motions with music, ultimately introducing the InterDance dataset. Visualizations of our dataset samples and the distribution of dance genres are shown in Figure 2. Statistical information about the dataset is presented in Table 1.

### 3.1 THE FEATURES OF THE INTERDANCE DATASET

The proposed InterDance dataset has the following key features: **Large-scale Data:** InterDance contains 3.93 hours of duet dance data, providing a substantial amount of training data to improve model generalization and reduce overfitting. To the best of our knowledge, this is currently the largest duet dance dataset. **Strong Interactions:** This dataset emphasizes strong interactions, such as handshakes, waist holds, which are essential elements for duet dance. **High Motion Quality:** This dataset is collected using state-of-the-art professional MoCap equipment for precise body and finger movements, with manual review to ensure the motion quality. **High Artistic quality:** We invite professional dancers with over a year of experience to perform their familiar dances. **Diverse Dance Genres:** Among duet dance datasets, InterDance includes the widest variety of 15 dance genres. **Long Duration per Sample:** InterDance reach the longest average sample duration ($\bar{T}$/s) of 142.7 seconds among duet dance datasets. Dances with longer durations contain rich and complete choreographic patterns.

### 3.2 THE DETAILS OF CONSTRUCTING THE DATASET

**MoCap Equipment:** The MoCap system we used consists of 16 infrared optical motion capture cameras, with a resolution of 2048x1536 and a frame rate of 120 fps. The subjects wore tight-fitting suits with 53 markers attached. The effective capture space is 7m x 7m x 3m. Due to the close articulation and occlusion issues of hand movements, the hand motions were captured using inertial data gloves.

Table 1: Comparisons of Dance Datasets. HHI means Human-Human Interaction, where 'W' means Weak Interaction without close contact, and 'S' means Strong Interaction otherwise. Among duet dance datasets with strong interaction, InterDance features the widest range of 15 dance genres, the longest average duration per sample at 142.7 seconds, and the total duration of 3.93 hours.

| Datasets | HHI | Music | Hand | Genres | $\bar{T}$/s | $T$ | Mocap | Representation |
|---|---|---|---|---|---|---|---|---|
| Dancing2Music (Lee et al., 2019) | × | ✓ | | 3 | 6 | 71.00h | | 2D Joints |
| DanceNet (Li et al., 2022) | × | ✓ | ✓ | 2 | - | 0.96h | ✓ | 3D Joints |
| PMSD (Valle-Pérez et al., 2021) | × | ✓ | ✓ | 3 | - | 3.10h | ✓ | 3D Joints |
| SMVR (Valle-Pérez et al., 2021) | × | ✓ | | 2 | - | 9.00h | | 3D Joints |
| AIST++ (Li et al., 2021b) | × | ✓ | | 10 | 13 | 5.20h | | SMPL |
| Motorica Dance(Alexanderson et al., 2023) | × | ✓ | ✓ | 8 | - | 6.22h | ✓ | BVH |
| FineDance (Li et al., 2023b) | × | ✓ | ✓ | 22 | 152.3 | 14.60h | ✓ | SMPL-X |
| AIOZ-GDANCE (Le et al., 2023b) | W | ✓ | | 7 | 37.5 | 16.70h | | SMPL |
| InterHuman (Liang et al., 2024) | S | | | n/a | 3.9 | 6.56h | | SMPL |
| DD100 (Siyao et al., 2024) | S | ✓ | ✓ | 10 | 69.3 | 1.92h | ✓ | SMPL-X |
| InterDance | S | ✓ | ✓ | 15 | 142.7 | 3.93h | ✓ | SMPL-X |

**Dataset Collection:** We invited professional dancers to perform duet dances based on music and used motion capture equipment to record their body and hand movements. To prevent foot floating issues, we required the dancers to wear flat shoes. The raw motion captured by MoCap was converted into SMPL-X (Pavlakos et al., 2019) format and manually inspected. For any problematic data, the dancers were invited to re-record the performance.

**Post process:** The MoCap system outputs the 3D coordinates of 53 markers on the body surface. After processing with the MoCap system software, it also provides the 3D coordinates of 55 internal human joints, including those of both the body and hands. Since the 3D coordinates of the 53 markers on the body surface contain detailed body surface information, we first used the SOMA (Ghorbani & Black, 2021) method to get the SMPL-X body shape parameters and body pose parameters from the markers. Next, we fixed the body shape parameters and used the 55 joint coordinates to optimize the poses for the body and hands, similar to Mosh++ Mahmood et al., 2019. Since the body and fingers operate in different motion spaces and have different data scales, we optimized the body and finger poses independently. We refined the body poses using 22 body joints and adjusted the finger poses based on the relative distances of 30 finger joints to the wrist. Finally, we combined the body and hand poses to obtain the human motion in SMPL-X format. The test and validation sets were randomly sampled proportionally from various dance genres, comprising 16.22% and 6.25% of the total data, respectively.

## 4 METHODOLOGY

Given music and the leader's dance $x^l$ as input, our goal is to generate the follower's dance $x^f$. For the given music, we follow (Li et al., 2021a) and employ Librosa (McFee et al., 2015) to extract the 2D music feature map $m \in \mathbb{R}^{T \times 35}$, where $T$ is the time length, 35 is the music feature channels with 1-dim envelope, 20-dim MFCC, 12-dim chroma, 1-dim one-hot peaks, and 1-dim one-hot beats. For the dance, we represent it as $x \in \mathbb{R}^{T \times C}$, where $C$ is the number of channels, details in the following chapter. Our method aims to model $p_\theta(x^f | m, x^l)$. The main challenge lies in the accuracy and physical interaction in the generated dance. We argue that previous methods encounter these issues due to the use of inappropriate motion representations and the lack of explicit fine-grained optimization for interactive movements during generation. To address these problems, we propose a new motion representation suitable for reactive motion generation and introduce a baseline generation algorithm based on diffusion. To enhance the physical realism of interactive movements, we also propose contact diffusion guidance and penetration diffusion guidance to optimize the quality of these interactions.

### 4.1 THE MOTION REPRESENTATION

As shown in Figure 3, existing methods use different motion representations to describe movements. SMPL-X (Pavlakos et al., 2019) parametrized body model achieves tremendous success. Through

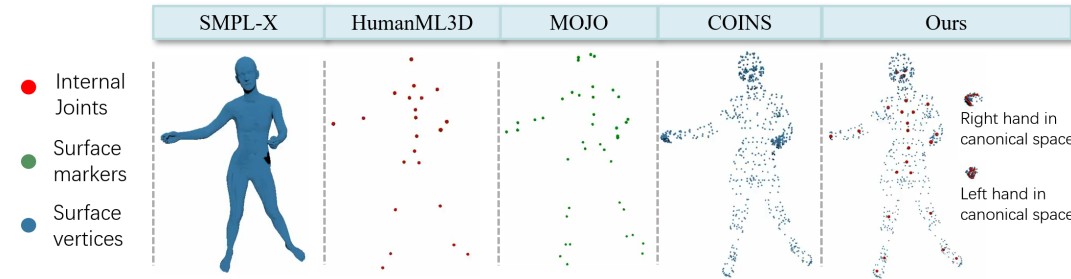

Figure 3: An overview of different motion representations.

Table 2: Comparisons of 3D motion representations.

| | SMPL-X (Pavlakos et al., 2019) | HumanML3D (Guo et al., 2022) | MOJO (Zhang et al., 2021) | COINS (Zhao et al., 2022) | Ours |
|---|---|---|---|---|---|
| Position space | | ✓ | ✓ | ✓ | ✓ |
| Joints | ✓ | ✓ | | | ✓ |
| Hands info | ✓ | | | ✓ | ✓ |
| Surface info | ✓ | | ✓ | ✓ | ✓ |
| Canonical space | ✓ | ✓ | | | ✓ |

the linear skinning function of SMPL-X, we can calculate the surface vertex positions of the human body with different shapes and movements. This allows for further optimization of strong interactions based on body surface vertices. However, the linear skinning function introduces huge computations and requires careful balancing of the weights of different losses, such as SMPL-X parameters and vertex positions. Additionally, existing work (Zhang et al., 2021) finds that the relative angle rotations propagated by the skeletal tree of SMPL-X exhibit highly nonlinear characteristics, which are not conducive to optimization in neural networks and can easily lead to foot-skating issues.

Recently, HumanML3D (Guo et al., 2022) proposes a motion representation based on the positions of 22 body joints. By operating in linear space and decoupling the trajectory and yaw angle, it is well-suited for single-person motion generation tasks. However, this method lacks finger and body surface information, making it difficult to use for fine-grained interactive motion generation tasks. MOJO (Zhang et al., 2021) and COINS (Zhao et al., 2022) utilize sparse body markers and vertices to represent motion, enabling accurate descriptions of both body shape and pose. This approach has shown great effectiveness in tasks of Human Scene Interaction and Human Object Interaction. However, these types of motion representations do not transform human motions into canonical space and lack motion contact labels, resulting in inadequate performance in Reactive Dance Generation tasks.

We find that while body vertices can accurately represent human shape and pose, incorporating internal body joints is more helpful. Based on this, contact labels can be added to enhance the accuracy of interactive motion, and transforming motion into canonical space can further increase the network's robustness and avoid overfitting. Based on the above observation, we propose a new motion representation. For the one person motion $\boldsymbol{x} = \{\boldsymbol{x}^i\}_{i=1}^{T}$, where $T$ is the frame number. The $\boldsymbol{x}^i$ is represented as:

$$\boldsymbol{x}^i = [r^x, r^y, r^z, r^a, \dot{r}^a, \dot{r}^x, \dot{r}^z, \boldsymbol{j}, \boldsymbol{v}, \dot{\boldsymbol{j}}, \dot{\boldsymbol{v}}, \boldsymbol{c}^{foot}, \boldsymbol{c}^p] \tag{1}$$

where $r^x, r^y, r^z$ is the root trajectory, $r^a$ is root angular along Y-axis (yaw angle), $\dot{r}^a$ is root angular velocity along Y-axis, $\dot{r}^x, \dot{r}^z$ is root linear velocities on the floor. $\boldsymbol{j}$ is the relative distances of the internal 55 joints. $\boldsymbol{v}$ is the relative distances of the surface 655 vertices, which is sampled following COINS (Zhao et al., 2022). For the body parts, we measure their distances from the root node, while for the fingers, we record their distances from the wrist. $\dot{\boldsymbol{j}}$ is joints velocities, $\dot{\boldsymbol{v}}$ is vertices velocities, $\boldsymbol{c}^{foot} \in \mathbb{R}^4$ is binary foot-ground contact label, $\boldsymbol{c}^p \in \mathbb{R}^{55+655}$ is the binary contact label of joints and vertices, $\boldsymbol{c}^p$ is set to 1 when the corresponding joint or vertex is in contact with another person.

## 4.2 THE DIFFUSION MODEL FOR REACT DANCE GENERATION

The diffusion model shows great potential in motion generation tasks (Tevet et al., 2022; Zhang et al., 2024; Zhou et al., 2023; Rempe et al., 2023; Li et al., 2023a; Tanaka & Fujiwara, 2023; Xu et al., 2023; Diller & Dai, 2024; Chen et al., 2024; Xu et al., 2024b; Karunratanakul et al., 2024). The most

direct approach to modeling $p_\theta(\boldsymbol{x}^f|\boldsymbol{m}, \boldsymbol{x}^l)$ is to use a conditional diffusion model. However, using the leader's motion as conditional input and solely relying on learned network parameters to generate follower motion does not guarantee the physical realism of the interaction between the leader and follower. To address this issue, we further propose an Interaction Refine Guidance sampling strategy to encourage accurate contact and avoid penetration. The overview of our method is shown in Figure 4. Given music feature $\boldsymbol{m}$, leader dance $\boldsymbol{x}^l$, Diffusion Time Step $N$ and sampled noise $\boldsymbol{x}_N^f$ as input, the Denoise Nework predict $\hat{\boldsymbol{x}}^f$ at each diffusion step. Then, we use the Interaction Refine Guidance to refine $\hat{\boldsymbol{x}}^f$ to $\tilde{\boldsymbol{x}}^f$. Next, we add noise to $\tilde{\boldsymbol{x}}^f$ and diffuse it to $\boldsymbol{x}_{N-1}^f$. After repeating this process for $N$ times, we can obtain the final predicted $\hat{\boldsymbol{x}}^f$ as output.

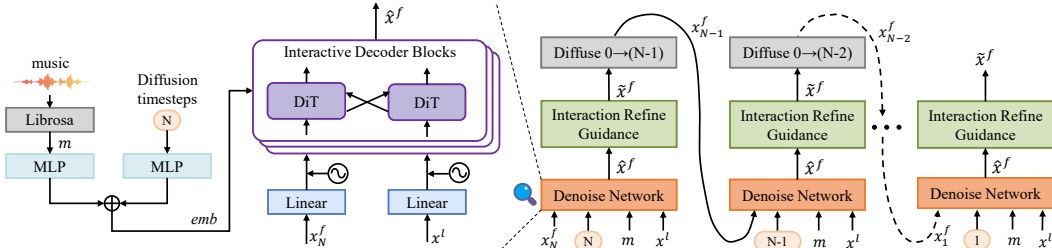

Figure 4: The right part shows our entire network, while the left part details the Denoise Network.

**Interaction Diffusion Model.** We utilize a conditional diffusion model (Ho et al., 2020) as our backbone. The diffusion model consists of a forward diffusion process that progressively adds noise to the clean data and a reverse diffusion process which is trained to reverse this process. The forward diffusion process introduces noise for $N$ steps formulated using a Markov chain:

$$q(\boldsymbol{x}_n^f|\boldsymbol{x}_{n-1}^f) := \mathcal{N}(\boldsymbol{x}_n^f; \sqrt{1-\beta_n}\boldsymbol{x}_{n-1}^f, \beta_n\boldsymbol{I}), \tag{2}$$

$$q(\boldsymbol{x}_{1:N}^f|\boldsymbol{x}_0^f) := \prod_{n=1}^{N} q(\boldsymbol{x}_n^f|\boldsymbol{x}_{n-1}^f), \tag{3}$$

where $\beta_n$ represents a fixed variance schedule and $\boldsymbol{I}$ is an identity matrix. The reverse diffusion process employs a learnable Denoise Network $f_\theta$ to denoise gradually. To facilitate the inclusion of loss functions for optimizing interactive motion during training, we directly predict clean data $\hat{\boldsymbol{x}}^f$ (Tevet et al., 2022; Liang et al., 2024), which can be formulated as:

$$\mathcal{L}_{recon} = \mathbb{E}_{\boldsymbol{x}^f, n}||f_\theta(\boldsymbol{x}_n^f, n, \boldsymbol{m}, \boldsymbol{x}^l) - \boldsymbol{x}^f||_2^2. \tag{4}$$

The Diffusion Transformer (DiT) shows excellent performance in image, video, and motion generation tasks (Peebles & Xie, 2023; Liu et al., 2024; Liang et al., 2024). Thus, we use DiT to construct the Interactive Decoder block, as detailed in the supplementary material.

**Auxiliary Losses.** We introduce several auxiliary losses to enhance training stability and physical realism like previous works (Liang et al., 2024). We recover the global position $\boldsymbol{p}^l \in \mathbb{R}^{T \times (55+655) \times 3}, \boldsymbol{p}^f \in \mathbb{R}^{T \times (55+655) \times 3}$ of human joints and vertices from the proposed canonical motion representation $\boldsymbol{x}^l, \boldsymbol{x}^f$, where 55 is the number of human joints, 655 is the number of downsampled human surface vertices. Then, to improve the smoothness of the generated dance, we add the velocity loss $\mathcal{L}_{vel}$ and acceleration loss $\mathcal{L}_{acc}$:

$$\mathcal{L}_{vel} = \left\|\boldsymbol{p}_{vel}^f - \hat{\boldsymbol{p}}_{vel}^f\right\|_2^2, \mathcal{L}_{acc} = \left\|\boldsymbol{p}_{acc}^f - \hat{\boldsymbol{p}}_{acc}^f\right\|_2^2, \tag{5}$$

where $\boldsymbol{p}_{vel}^f, \boldsymbol{p}_{acc}^f$ are the velocity and acceleration computed from $\boldsymbol{p}^f$. To optimize the quality of foot-ground contact, we further add a foot contact loss $\mathcal{L}_{foot} = \left\|\hat{\boldsymbol{p}}_{vel}^{foot} \odot \hat{\boldsymbol{c}}^{foot}\right\|$, where $\hat{\boldsymbol{p}}_{vel}^{foot}$ is the corresponding foot joints velocity of generated follower motion. Inspired by InterGen (Liang et al., 2024), to optimize the physical realism of interactive motion, we add the Distance Matrix loss $\mathcal{L}_{DM}$ and Relative Orientation loss $\mathcal{L}_{RO}$ as follows:

$$\mathcal{L}_{DM} = \left\|(M(\boldsymbol{p}^l, \hat{\boldsymbol{p}}^f) - M(\boldsymbol{p}^l, \boldsymbol{p}^f)) \odot I(M(\boldsymbol{p}^l, \boldsymbol{p}^f) < \bar{M})\right\|_2^2,$$
$$\mathcal{L}_{RO} = ||O(\boldsymbol{x}^l, \hat{\boldsymbol{x}}^f) - O(\boldsymbol{x}^l, \boldsymbol{x}^f)||_2^2, \tag{6}$$

where $M$ denotes the joint distance map of the leader and follower, and $I$ is an indicator function used to mask the loss. This loss is activated only when the distance between the two people is less than the distance threshold $\bar{M}$. $\odot$ indicates Hadamard product. $O$ indicates the relative angle between the two dancers around the Y-axis. We also introduce a contact loss $\mathcal{L}_{con}$, utilizing contact labels to encourage accurate interaction between two dancers,

$$\mathcal{L}_{con} = ||M_{min}(\boldsymbol{p}^l, \hat{\boldsymbol{p}}^f) \odot \boldsymbol{c}^{pl}||_2^2 + ||M_{min}(\hat{\boldsymbol{p}}^f, \boldsymbol{p}^l) \odot \hat{\boldsymbol{c}}^{pf}||_2^2, \tag{7}$$

where $\{M_{min}(\boldsymbol{p}^l, \hat{\boldsymbol{p}}^f)\}_i$ denotes the shortest distance from $\{\boldsymbol{p}^l\}_i$ to $\hat{\boldsymbol{p}}^f$. $\boldsymbol{c}^{pl}, \hat{\boldsymbol{c}}^{pf}$ represents the contact label in the motion representation of $\boldsymbol{x}^l, \hat{\boldsymbol{x}}^f$. These loss functions help the network learn the interactive motion between dancers. Our overall training object is the weighted sum of these losses:

$$\mathcal{L}_{total} = \mathcal{L}_{recon} + \lambda_{vel}\mathcal{L}_{vel} + \lambda_{acc}\mathcal{L}_{acc} + \lambda_{DM}\mathcal{L}_{DM} + \lambda_{RO}\mathcal{L}_{RO} + \lambda_{con}\mathcal{L}_{con} + \lambda_{foot}\mathcal{L}_{foot}. \tag{8}$$

**Interaction Refine Guidance.** We find that the above conditional diffusion model generates follower dances that perform well in terms of motion smoothness and self-motion realism, and can also roughly interact with the leader. However, there are still many artifacts, such as contact floating or penetration, especially in strong interactive movements. Therefore, we propose a diffusion-based sampling guidance strategy that calculates contact guidance gradients $\nabla\mathcal{L}_{con}$ and penetration guidance gradients $\nabla\mathcal{G}_{pene}$. Then we use these gradients to guide the denoising process at each diffusion step, achieving more accurate contact and suppressing penetration. $\mathcal{L}_{con}$ is also the contact optimization loss function formulated as Equ. 7. Its gradient can be used to guide the denoising process to avoid contact floating issues explicitly. To tackle the penetration problems, we compute the Signed Distance Field (SDF) between the follower's surface vertices and the leader's mesh using the COAP (Mihajlovic et al., 2022; Zhang et al., 2023) model. By querying each joint and surface vertex of the follower, the penetration cost function is defined as:

$$\mathcal{G}_{pene}(\boldsymbol{x}_n^f) = \frac{1}{|\boldsymbol{p}_n^f|} \sum_{q \in \boldsymbol{p}_n^f} \sigma\big(f_\Theta(q|\mathcal{S})\big)\mathbb{I}_{f_\Theta(\boldsymbol{p}_n^f|\mathcal{S})>0} \tag{9}$$

where $\boldsymbol{p}_n^f$ is the point set recovered from $\boldsymbol{x_n^f}$ and $f_\Theta(q|\mathcal{S})$ is the COAP model used to compute the signed distance between point $q$ and the leader mesh $\mathcal{S}$, $\sigma(\cdot)$ stands for the sigmoid function. At each denoise step, the gradient of $\mathcal{L}_{con}$ and $\mathcal{G}_{pene}$ are utilized to guide the diffusion denoising process:

$$\widetilde{\boldsymbol{x}}_n^f = \hat{\boldsymbol{x}}_n^f + a_{con}\nabla\mathcal{L}_{con}(\hat{\boldsymbol{x}}_n^f) + a_{pene}\nabla\mathcal{G}_{pene}(\hat{\boldsymbol{x}}_n^f) \tag{10}$$

where $a_{con}$ and $a_{pene}$ are scale factors and $\widetilde{\boldsymbol{x}}_n^f$ is the motion representation after guidance.

## 5 EXPERIMENTS

**Implementation Details.** The initial learning rate is 1e-4, with a weight decay of 2e-5. The model trains for 1000 epochs. The number of heads in multi-head attention is uniformly set to 8. In the ablation study, all variants are trained for 500 epochs under the same experimental settings, except for the conditions being investigated. The training is conducted on 4 NVIDIA 4090 GPUs, taking approximately 22 hours in total. The inference time averages 1.6 seconds with a batch size of 8.

**Evaluation Metrics.** 1) **Frechet Inception Distance (FID).** We use FID to measure the degree of closeness between the generated motion of the follower and the ground truth. 2) **Diversity (Div).** We use Div to assess the average feature distance of generated dances. FID and Div are calculated on kinematic (Onuma et al., 2008) and graphical (Müller et al., 2005) features, and described as $\text{FID}_k, \text{FID}_g, \text{Div}_k, \text{Div}_g$ respectively. 3) **Cross Distance (cd).** Following Duolando(Siyao et al., 2024), we compute pairwise distances between ten joints of the leader and follower (including the pelvis, knees, feet, shoulders, head, and wrists) as interaction features, and use these features to calculate $\text{FID}_{cd}$ and $\text{Div}_{cd}$. 4) **Contact Frequency (CF).** CF represents the ratio of contact frames to all frames. We define contact as vertices distances between dancers being less than 1cm. 5) **Penetration Rate (PR).** We use the ratio of penetration vertices to all vertices as the Penetration Rate. 6) **Beat Echo Degree (BED).** Following Duolando(Siyao et al., 2024), we use BED to measure the rhythm consistency between two dancers. 7) **Beat-Align Score (BAS).** We follow AIST++(Li et al., 2021a) and use BAS to assess the rhythm matching between music and dance.

Table 3: Comparisons of reactive dance generation, all methods train&test on the InterDance dataset.

| Method | Motion Quality | | | | Interaction Quality | | | | | Rhythmic |
|---|---|---|---|---|---|---|---|---|---|---|
| | $FID_k(\downarrow)$ | $FID_g(\downarrow)$ | $Div_k(\uparrow)$ | $Div_g(\uparrow)$ | $FID_{cd}(\downarrow)$ | $Div_{cd}(\uparrow)$ | $PR(\downarrow)$ | $CF(\uparrow)$ | $BED(\uparrow)$ | $BAS(\uparrow)$ |
| Ground Truth | 10.41 | 2.17 | 15.47 | 7.38 | 1.4 | 10.94 | 0.55% | 9.91% | 0.3404 | 0.1882 |
| EDGE (Tseng et al., 2023) | 55751.37 | 37.7 | **13.74** | 4.08 | 5.62 | 10.79 | **0.12%** | 2.15% | 0.2524 | **0.2309** |
| GCD (Le et al., 2023a) | 202.12 | 19.34 | 6.16 | 4.70 | 4.06 | **11.34** | 0.28% | 2.23% | 0.2432 | 0.2025 |
| InterGen (Liang et al., 2024) | 114.39 | 12.05 | 8.82 | 5.74 | 1.79 | 11.29 | 0.41% | 5.55% | 0.3506 | 0.1987 |
| Duolando (Siyao et al., 2024) | 86.79 | 8.14 | 10.13 | **6.07** | 4.19 | 9.78 | 0.19% | 3.49% | 0.2608 | 0.1995 |
| Ours | **65.96** | **8.02** | 10.53 | **6.07** | **1.51** | 10.89 | 0.36% | **6.99%** | **0.3644** | 0.1992 |

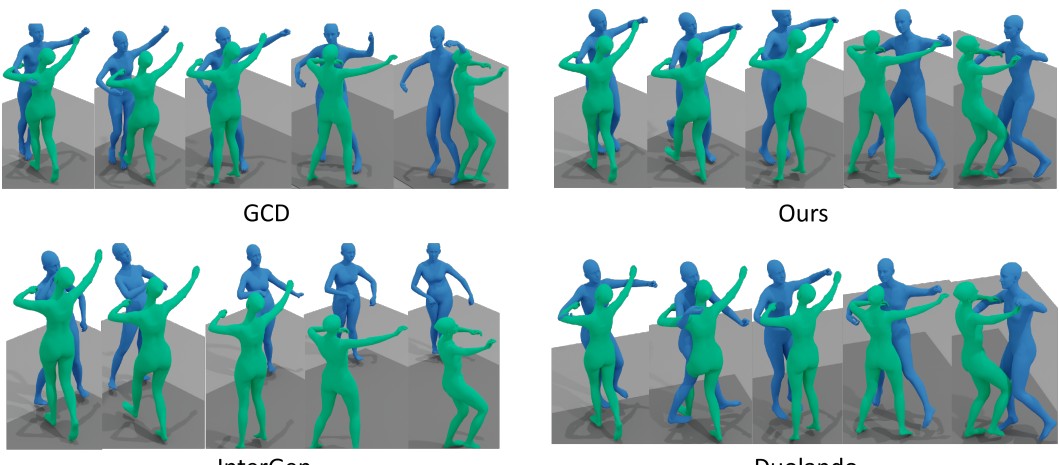

GCD        Ours

InterGen        Duolando

Figure 5: Qualitative comparisons of reactive dance generation, blue is the generated follower.

## 5.1 QUANTITATIVE AND QUALITATIVE EVALUATION

**Quantitative Comparisons.** We compare our method with several state-of-the-art approaches from related fields on our InterDance dataset, all of which are adapted for benchmarking in the context of react dance generation. In the field of solo dance generation, we choose EDGE (Tseng et al., 2023) for comparison. For group dance generation, we select the latest GCD (Le et al., 2023a) for comparison. For text-driven human interaction generation, we select InterGen (Liang et al., 2024). We also evaluate against Duolando (Siyao et al., 2024), currently the only available method for generating react dance. As shown in Table 3, our method demonstrates superior performance compared with previous approaches, particularly in quality-related FID and Div metrics. In terms of contact metrics CF, our method achieves a significant improvement of 4.84% compared to EDGE, demonstrating our advantages in interaction generation. Although the motions generated by EDGE and GCD have low PR metric, their CF remains poor. The text-driven interactive generation method, InterGen, performs well on interaction quality but relatively poorly on motion quality due to the variation in signal modalities. Compared with Duolando, which is carefully designed for reactive dance generation, our method performs better in most metrics. This is because Duolando employs a two-stage training framework that stores motions in a codebook, making it difficult to optimize fine-grained interactions. In contrast, our method explicitly optimizes interactions within the original motion space during sampling.

**Qualitative Comparisons.** We provide visualizations of the results generated by different methods in Figure 5. This figure demonstrates that while GCD (Le et al., 2023a) initially generates relatively good follower motions, it later exhibits shifts in the relative positions of the dancers. InterGen (Liang et al., 2024) produces monotonous motions that fail to appropriately respond to the leader. Although the results of Duolando (Siyao et al., 2024) are relatively good, it also suffers from significant contact floating or penetration issues. In contrast, our method produces more responsive and interactively coherent motions, achieving superior contact quality and mitigating penetration problems.

**User Study.** To obtain subjective evaluations of the generated results from different methods, we recruit 40 participants for a user study. Each participant watches 40 pairs of videos with 30 fps 1280x1024 resolution. One video generated by our method and the other by either the ground truth or another method. Among these 40 participants, 26 have no dance background, while 14 have a dance background. We randomize the order of the dances to ensure fairness. We ask participants the following questions: *Which dance has better overall motion quality? Which dance has better interaction? Which dance has better coordination with the music?* As depicted in Figure 6, our method outperforms all other methods in more than 85% of cases. This further demonstrates the effectiveness of our approach.

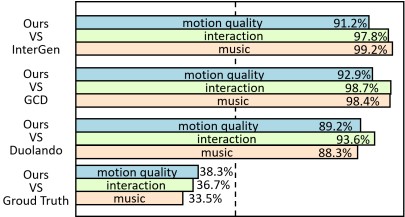

Figure 6: Reactive dance generation user study.

## 5.2 Ablation Study

We conduct ablation studies on the reactive dance generation task using the InterDance dataset.

**Motion Representation.** To validate the effectiveness of our proposed motion representation in generating human interactive dances, we conduct a series of experiments. 1) **Joints** denotes the normal motion representation of 55 human joints and 2) **Vertices** indicates adding 655 surface points to the motion representation. 3) **Canonical** means transforming motion into canonical space. The corresponding experimental results are shown in Table 4. Generating dance using only 55 **joints** results in good motion quality but performs poorly on interaction quality. This is because interaction requires finer perceptual granularity, and using only 55 joints loses much critical information about the human surface. Therefore, we addressed this issue by adding surface **vertices** points to the motion representation, which greatly improved the $\text{FID}_{cd}$ from 26.73 to 8.11 and lowered the PR. Although this indeed improved the quality of interaction, the additional 655 vertices also increased the difficulty of dance modeling, leading to a decrease in motion quality. To resolve this, we further processed the joints and vertices into **canonical** space to simplify dance modeling, ultimately achieving good results in both motion quality and interaction quality.

Table 4: Effect of the proposed motion representation.

| Joints | Vertices | Canonical | Motion Quality | | | | Interaction Quality | | | | |
|---|---|---|---|---|---|---|---|---|---|---|---|
| | | | $\text{FID}_k(\downarrow)$ | $\text{FID}_g(\downarrow)$ | $\text{Div}_k(\uparrow)$ | $\text{Div}_g(\uparrow)$ | $\text{FID}_{cd}(\downarrow)$ | $\text{Div}_{cd}(\uparrow)$ | $\text{PR}(\downarrow)$ | $\text{CF}(\uparrow)$ | $\text{BED}(\uparrow)$ |
| ✓ | | | **83.75** | 8.51 | 10.43 | 6.32 | 26.73 | 8.17 | 0.78% | **8.33%** | 0.3699 |
| ✓ | ✓ | | 102.26 | 10.16 | **11.27** | 6.21 | 10.94 | **13.17** | **0.36%** | 4.21% | 0.3692 |
| ✓ | ✓ | ✓ | 88.82 | **8.41** | 10.74 | **6.36** | **8.11** | 10.45 | **0.36%** | 5.98% | **0.3961** |

**Interaction Losses.** We evaluate the impact of different components of the interaction loss function on the model performance. The results in Table 5 show that using only $\mathcal{L}_{DM}$ can yield good generation results, but it performs poorly on $\text{FID}_{cd}$. This is because the $\mathcal{L}_{DM}$ only calculates distances and has a weak perception of angles. Therefore, we include the $\mathcal{L}_{RO}$, which constrains the relative trajectories of dancers in vector form, resulting in a significant improvement in $\text{FID}_{cd}$. After adding the $\mathcal{L}_{contact}$, both motion quality and interaction quality are greatly improved, as contact loss directly encourages reasonable interaction between dancers.

Table 5: Effect of Interaction losses.

| $\mathcal{L}_{DM}$ | $\mathcal{L}_{RO}$ | $\mathcal{L}_{contact}$ | Motion Quality | | | | Interaction Quality | | | | |
|---|---|---|---|---|---|---|---|---|---|---|---|
| | | | $\text{FID}_k(\downarrow)$ | $\text{FID}_g(\downarrow)$ | $\text{Div}_k(\uparrow)$ | $\text{Div}_g(\uparrow)$ | $\text{FID}_{cd}(\downarrow)$ | $\text{Div}_{cd}(\uparrow)$ | $\text{PR}(\downarrow)$ | $\text{CF}(\uparrow)$ | $\text{BED}(\uparrow)$ |
| ✓ | | | 86.71 | 8.31 | 10.69 | **6.44** | 8.55 | 10.35 | 0.37% | 5.61% | 0.3664 |
| ✓ | ✓ | | 94.27 | 9.65 | 10.53 | 6.41 | 5.61 | 10.43 | **0.36%** | 5.74% | 0.3603 |
| ✓ | ✓ | ✓ | **78.65** | **8.19** | **10.75** | 6.38 | **4.49** | **10.61** | 0.39% | **6.62%** | **0.3831** |

**Interaction Refine Guidance.** To assess the effectiveness of our diffusion guidance sampling strategy, we set four distinct settings to do ablation studies as Table 6 shows. The results indicate that **contact** guidance can facilitate accurate contact between dancers and improve the interaction quality, but it is ineffective in addressing penetration issues. **Penetration** suppression guidance can suppress penetration occurrences, reducing penetration rate from 0.45% to 0.35%, but it may also lead to a

Table 6: Effect of the Interaction Refine Guidance.

| contact-guide | pene-guide | Motion Quality | | | | Interaction Quality | | | | |
|---|---|---|---|---|---|---|---|---|---|---|
| | | $\text{FID}_k(\downarrow)$ | $\text{FID}_g(\downarrow)$ | $\text{Div}_k(\uparrow)$ | $\text{Div}_g(\uparrow)$ | $\text{FID}_{cd}(\downarrow)$ | $\text{Div}_{cd}(\uparrow)$ | $\text{PR}(\downarrow)$ | $\text{CF}(\uparrow)$ | $\text{BED}(\uparrow)$ |
| | | 88.25 | 9.85 | 10.61 | **6.49** | 9.55 | 10.35 | 0.43% | 6.07% | 0.3803 |
| ✓ | | 87.52 | **9.05** | 10.63 | 6.48 | 9.02 | 10.46 | 0.45% | **6.33**% | **0.3885** |
| | ✓ | 86.41 | 9.17 | 10.60 | 6.43 | 9.18 | 10.47 | 0.35% | 5.91% | 0.3822 |
| ✓ | ✓ | **86.18** | 9.07 | **10.66** | 6.47 | **8.74** | **10.51** | **0.34%** | 6.12% | 0.3843 |

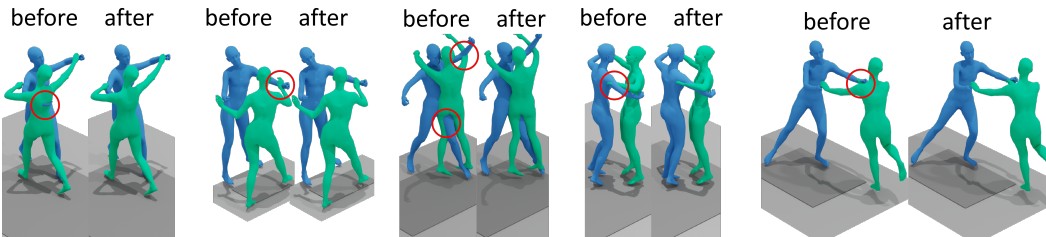

Figure 7: The visual comparison of the effects before and after using the diffusion guidance strategy.

decrease in some interaction metrics. The coordinated use of **both** can minimize the occurrence of penetration phenomena while ensuring the quality of interaction, thereby enhancing dance realism.

## 5.3 DUET DANCE GENERATION

The task of duet dance generation is to generate a two-person dance from given music. Directly generating duet dances has lower interaction quality due to the lack of leader guidance. By replacing the input leader dance $x^l$ with sampled noise $x^l_N$ and requiring the denoise network to predict both $\hat{x}^l$ and $\hat{x}^f$, our method can achieve duet dance generation. Table 7 shows that our method improves both motion and interaction quality over other approaches. All methods are trained on the InterDance train set. Please refer to the supplementary material for more details.

Table 7: Quantitative comparisons of duet dance generation on the InterDance dataset.

| Method | Motion Quality | | | | Interaction Quality | | | | | Rhythmic |
|---|---|---|---|---|---|---|---|---|---|---|
| | $\text{FID}_k(\downarrow)$ | $\text{FID}_g(\downarrow)$ | $\text{Div}_k(\uparrow)$ | $\text{Div}_g(\uparrow)$ | $\text{FID}_{cd}(\downarrow)$ | $\text{Div}_{cd}(\uparrow)$ | $\text{PR}(\downarrow)$ | $\text{CF}(\uparrow)$ | $\text{BED}(\uparrow)$ | $\text{BAS}(\uparrow)$ |
| Ground Truth | 7.42 | 1.21 | 15.32 | 7.20 | 1.40 | 10.94 | 0.54% | 9.91% | 0.3471 | 0.1901 |
| EDGE (Tseng et al., 2023) | 2566.12 | 67.81 | **36.52** | 5.78 | 110.16 | 4.21 | 1.25% | 3.03% | 0.2503 | **0.2189** |
| GCD (Le et al., 2023a) | 81.24 | 14.63 | 9.86 | 5.26 | 43.46 | 8.28 | 0.65% | 6.06% | 0.2664 | 0.2045 |
| InterGen (Liang et al., 2024) | 107.96 | 14.06 | 8.54 | 4.83 | 73.37 | 7.96 | 0.44% | **6.56%** | 0.2451 | 0.2135 |
| Duolando (Siyao et al., 2024) | 77.29 | 10.37 | 10.25 | 5.64 | 110.82 | **12.99** | **0.09%** | 1.92% | 0.2416 | 0.2071 |
| Ours | **72.27** | **8.07** | 9.94 | **5.82** | **36.84** | 7.92 | 0.18% | 2.93% | **0.2677** | 0.1998 |

## 6 CONCLUSION AND LIMITATION

In this work, we introduce InterDance, which includes a large-scale, high-quality duet dance dataset and a reactive/duet dance generation method. To enhance the quality of the generated interactive dance, we train the diffusion model using our proposed new motion representation and optimize interaction quality with diffusion guidance techniques. Our method shows excellent results, as demonstrated by both quantitative and qualitative experiments. However, there are still **limitations**, such as the trade-off between quality and scale in our dataset, inefficient for generating long-sequence duet dances in our method. The potential societal impact is that, as dance generation and human interaction technology become more advanced, highly realistic virtual humans might lead users to become so immersed in the virtual world that they detach from real-world participation.

## 7 REPRODUCIBILITY STATEMENT

We have elucidated our design in the paper including the dataset construction process (Section 3), model structure (Section 4), and the training and testing details (Section 5). More details of the duet dance generation, interactive decoder, user study and instructions for dancers are in the appendix. To facilitate the reproduction, we will make our **code**, **dataset**, and **weights** publicly available.

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

# INTERDANCE: REACTIVE 3D DANCE GENERATION WITH REALISTIC DUET INTERACTIONS SUPPLEMENTARY MATERIAL

**Anonymous authors**

## CONTENTS

## A   MORE DETAILS OF INTERDANCE DATASET

As shown in Table 1, we summarize the dance data for different genres in terms of total duration (in minutes), number of frames, number of sequences, and their percentage of the entire InterDance dataset.

## B   MORE RETRIEVAL BASED SIMPLE BASELINES

As shown in Table 2, We add **NN-motion** and **NN-music** as expanded baselines following Ng et al. (2022). Since NN-motion and NN-music are directly retrieved from the training set, they exhibit high quantitative metrics. However, the simple retrieval ignores interactive motion, making it impossible for the generated follower dancer to perform alongside the leader. The follower's dance movements lack even basic interaction with the leader in cases. The calculation methods for NN motion and NN music metrics are as follows:

NN-motion: Given an input leader motion, we find its nearest neighbor from the training set and use its corresponding follower segment as the prediction. The distance is calculated based on the MSE distance of the motion clips.

NN-music: Given an input music clip on InterDance test set, we find its nearest neighbor music clip from the training set and use its corresponding follower motion as the prediction. The distance is

Table 1: More Details of InterDance Dataset

| Genre | Minutes | Frames | Sequence number | Percentage |
|---|---|---|---|---|
| Jive | 8.20 | 59006 | 3 | 3.48% |
| Chacha | 11.25 | 81000 | 4 | 4.78% |
| HanTang | 15.44 | 111154 | 6 | 6.56% |
| Kpop | 53.79 | 387262 | 21 | 22.84% |
| Jazz | 6.61 | 47,610 | 4 | 2.81% |
| Samba | 10.69 | 76982 | 6 | 4.54% |
| Waltz | 10.79 | 77,674 | 8 | 4.58% |
| Rumba | 27.11 | 195162 | 7 | 11.51% |
| Dai | 11.95 | 86034 | 6 | 5.07% |
| HipHop | 31.87 | 229432 | 10 | 13.53% |
| Wei | 11.78 | 84822 | 6 | 5.00% |
| ShenYun | 18.44 | 132758 | 8 | 7.83% |
| DunHuang | 6.41 | 46172 | 3 | 2.72% |
| Urban | 6.58 | 47390 | 4 | 2.80% |
| Miao | 4.58 | 32,978 | 3 | 1.95% |

calculated based on the MSE distance of the music features extracted by the advanced music feature extractor Jukebox (Dhariwal et al., 2020).

Table 2: Quantitative comparisons of duet dance generation on the InterDance dataset, with expanded retrieval based baseline of NN-motion and NN-music.

| Method | Motion Quality | | | | Interaction Quality | | | | | Rhythmic |
|---|---|---|---|---|---|---|---|---|---|---|
| | $\text{FID}_k(\downarrow)$ | $\text{FID}_g(\downarrow)$ | $\text{Div}_k(\uparrow)$ | $\text{Div}_g(\uparrow)$ | $\text{FID}_{cd}(\downarrow)$ | $\text{Div}_{cd}(\uparrow)$ | $\text{PR}(\downarrow)$ | $\text{CF}(\uparrow)$ | $\text{BED}(\uparrow)$ | $\text{BAS}(\uparrow)$ |
| Ground Truth | 7.42 | 1.21 | 15.32 | 7.20 | 1.40 | 10.94 | 0.54% | 9.91% | 0.3471 | 0.1901 |
| NN-motion | 48.4 | 7.17 | 13.22 | 8.19 | 2.07 | 10.81 | 0.52% | 5.54% | 0.2356 | 0.1904 |
| NN-music | 24.01 | 4.34 | 16.53 | 6.78 | 14.07 | 12.76 | 0.18% | 3.04% | 0.1966 | 0.1890 |
| EDGE (Tseng et al., 2023) | 2566.12 | 67.81 | **36.52** | 5.78 | 110.16 | 4.21 | 1.25% | 3.03% | 0.2503 | **0.2189** |
| GCD (Le et al., 2023) | 81.24 | 14.63 | 9.86 | 5.26 | 43.46 | 8.28 | 0.65% | 6.06% | 0.2664 | 0.2045 |
| InterGen (Liang et al., 2024) | 107.96 | 14.06 | 8.54 | 4.83 | 73.37 | 7.96 | 0.44% | **6.56%** | 0.2451 | 0.2135 |
| Duolando (Siyao et al., 2024) | 77.29 | 10.37 | 10.25 | 5.64 | 110.82 | **12.99** | **0.09%** | 1.92% | 0.2416 | 0.2071 |
| Ours | **72.27** | **8.07** | 9.94 | **5.82** | 36.84 | 7.92 | 0.18% | 2.93% | **0.2677** | 0.1998 |

## C MORE DETAILS OF DUET DANCE GENERATION

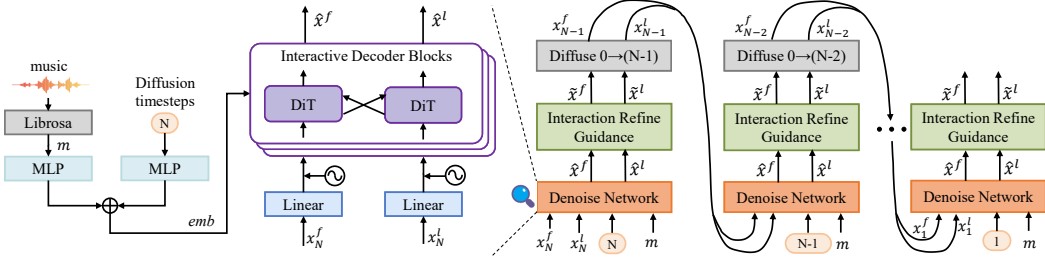

Figure 1: The architecture of our method's variant for duet dance generation.

Our method also supports generating duet dance directly from music without needing given leader dance $x^l$ as input. The duet dance generation task need to model $p_\theta(x^l, x^f|m)$. The overview of our method for duet dance generation is shown in Figure 1. Given music feature $m$, Diffusion Time Step $N$ and sampled noise $(x_N^l, x_N^f)$ as input, the Denoise Nework predict $(\hat{x}^l, \hat{x}^f)$ at each diffusion step. Then, we use the Interaction Refine Guidance to refine $(\hat{x}^l, \hat{x}^f)$ to $(\tilde{x}^l, \tilde{x}^f)$. Next, we add noise to $(\tilde{x}^l, \tilde{x}^f)$ and diffuse it to $(x_{N-1}^l, x_{N-1}^f)$. After repeating this process for N times,

we can obtain the final predicted $(\hat{\boldsymbol{x}}^l, \hat{\boldsymbol{x}}^f)$ as output. The diffusion reconstruction loss function can be formulated as:

$$\mathcal{L}_{recon} = \mathbb{E}_{(\boldsymbol{x}^l, \boldsymbol{x}^f, n)}||f_{\boldsymbol{\theta}}(\boldsymbol{x}_n^l, \boldsymbol{x}_n^f, n, \boldsymbol{m}) - (\boldsymbol{x}^l, \boldsymbol{x}^f)||_2^2. \tag{1}$$

For the auxiliary losses described in the main paper, simply replace $\boldsymbol{x}^l$ with the predicted $\hat{\boldsymbol{x}}^l$ to train the duet dance generation network.

## D  MORE DETAILS OF THE INTERACTIVE DECODER BLOCK

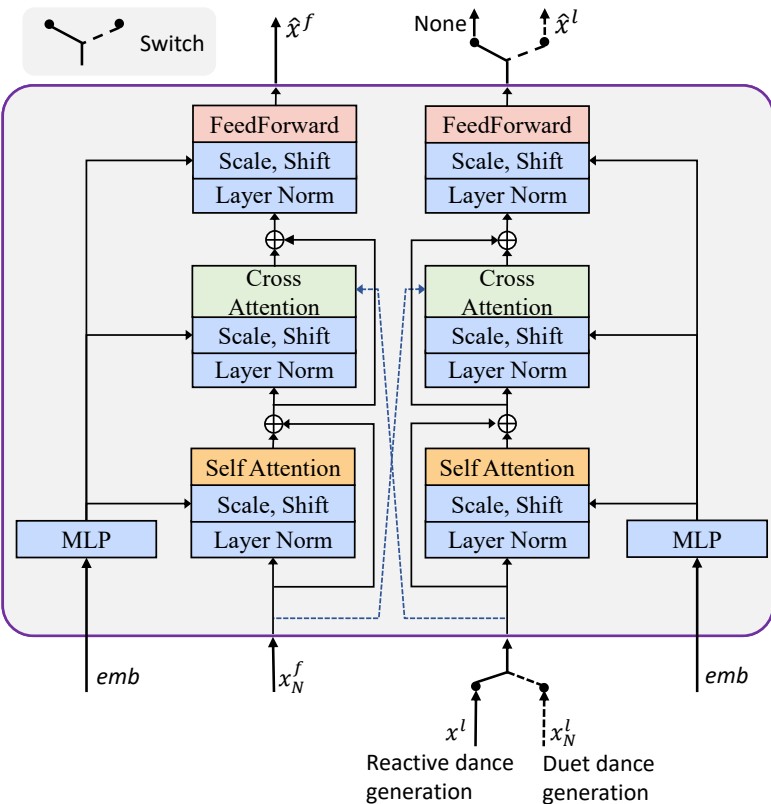

Figure 2: The detailed architecture of the Interactive Decoder Block. 'emb' is the embedding of music and diffusion time step.

The Interactive Decoder Block primarily consists of two Diffusion Transformers (DiT) and utilizes the cross-attention layer as the communication mechanism between the leader and the follower. The detailed network structure is shown in Figure 2. The Interactive Decoder Block also supports duet dance generation, which involves generating dances for two people directly based on music. For the duet dance generation task, we simply replace $x^l$ with $x_N^l$ and configure the Interactive Decoder Block to output both $\hat{x}^f$ and $\hat{x}^f$ simultaneously.

## E  INPUT SIGNAL ANALYSIS

Due to the react dance generation task involving both music and leader motion as inputs, we designed experiments to explore the role of different signals in dance generation. The experimental results are shown in Table 3. The **w/o. motion** and **w/o. music** represent training without using the leader's motion sequence or music signal, respectively. The **mask motion** and **mask music** represent masking the leader's motion sequence or music signal to zero during testing (the model is trained with both leader motion and music). The experimental results indicate that the leader's motion sequence plays a more critical role in interactive motion generation compared to background

music signals. Training the follower generation model based solely on music or blocking the leader's motion signals in a trained model both result in significant declines in motion quality and interaction quality. The impact of missing music signals is primarily reflected in the BAS metric, which is strongly related to music rhythm. Additionally, the absence of music signals also affects motion quality and interaction quality to some extent.

Table 3: Effect of input signals, tested on reactive dance generation.

| Method | Motion Quality | | | | Interaction Quality | | | | | Rhythmic |
|---|---|---|---|---|---|---|---|---|---|---|
| | $FID_k(\downarrow)$ | $FID_g(\downarrow)$ | $Div_k(\uparrow)$ | $Div_g(\uparrow)$ | $FID_{cd}(\downarrow)$ | $Div_{cd}(\uparrow)$ | $CF(\uparrow)$ | $PR(\downarrow)$ | $BED(\uparrow)$ | $BAS(\uparrow)$ |
| w/o. motion | 176.78 | 32.54 | 7.13 | 4.11 | 7.49 | **12.31** | 0.17% | 2.73% | 0.2397 | **0.2072** |
| w/o. music | 97.15 | 10.88 | 9.29 | 5.86 | 1.23 | 11.11 | **0.41**% | 6.42% | **0.3735** | 0.1821 |
| mask motion | 191.74 | 27.26 | 5.43 | 4.14 | 4.01 | 11.19 | 0.17% | **2.45**% | 0.2356 | 0.1963 |
| mask music | 79.95 | 9.81 | 10.05 | 6.01 | **0.99** | 11.19 | 0.35% | 5.59% | 0.3555 | 0.1803 |
| Ours | **65.96** | **8.02** | **10.53** | **6.07** | 1.51 | 10.89 | 0.36% | 6.99% | 0.3644 | 0.1992 |

# F MORE EXPERIMENTAL RESULTS OF INTERACTION QUALITY.

In the main paper, we employ the contact metric CF to represent the percentage of frames in which the distance between dancers is below a certain threshold. However, this metric is highly susceptible to outliers and weak interactions, preventing it from accurately reflecting the quality of contact. To address this issue, we propose three new metrics, contact leader rate (CLR), contact follower rate (CFR), and contact vertices rate (CVR), which calculate the proportion of vertices where the distance between dancers is below the threshold. This approach mitigates the influence of isolated points on the evaluation results. CLR represents the proportion of leader vertices in contact, while CVR is the average of CLR and CFR (the proportion of follower vertices in contact). The experimental results for the aforementioned metrics are displayed in Table 4. Our method exceeds all others regarding contact metrics, achieving more than twice the performance of Duolando, which is specifically designed for generating react dances.

# G EXTENDED EXPERIMENTS ON DD100 DATASET.

To validate the applicability of our method on the DD100 datasets and explore the potential benefits of the InterDance dataset for testing on DD100 datasets, we conducted a series of experiments. Although both DD100 and InterDance are in SMPL-X format, their coordinate systems differ. DD100 adopts the Blender coordinate system (z-axis up) for easier rendering and visualization, while InterDance uses the SMPL coordinate system (y-axis up) for better compatibility with other SMPL-based datasets commonly used in the academic community. To conduct the dataset ablation experiments and enable our method to train on both datasets simultaneously, we converted the motion data of DD100 into the SMPL coordinate system, which is consistent with InterDance. After this conversion, the ground truth metrics of DD100 also changed. Therefore, we remeasured the metrics on the test set of DD100, as show in the last line of Table 5. **InterDance** and **DD100** indicate whether the

Table 4: Detailed results of contact metrics for various methods under the reactive dance generation task, tested on our InterDance dataset.

| Method | $CF(\uparrow)$ | $CLR(\uparrow)$ | $CFR(\uparrow)$ | $CVR(\uparrow)$ |
|---|---|---|---|---|
| Ground Truth | 9.9051% | 0.0599% | 0.0667% | 0.0633% |
| GCD | 2.3293% | 0.0104% | 0.0095% | 0.0099% |
| EDGE | 2.1502% | 0.0076% | 0.0076% | 0.0076% |
| InterGen | 5.5506% | 0.0274% | 0.0229% | 0.0251% |
| Duolando | 3.4017% | 0.0101% | 0.0105% | 0.0103% |
| Ours | **6.9801%** | **0.0486%** | **0.0377%** | **0.0432%** |

Table 5: Experiments on the test set of DD100 Dataset. DD100 and InterDance indicate whether these datasets (train set) are used for training. Fine-tuning refers to pre-training on InterDance followed by fine-tuning on DD100.

| DD100 | InterDance | Fine-tuning | Motion Quality | | | | Interaction Quality | | | | |
|-------|------------|-------------|----------------|--|--|--|---------------------|--|--|--|--|
| | | | $FID_k(\downarrow)$ | $FID_g(\downarrow)$ | $Div_k(\uparrow)$ | $Div_g(\uparrow)$ | $FID_{cd}(\downarrow)$ | $Div_{cd}(\uparrow)$ | $PR(\downarrow)$ | $CF(\uparrow)$ | $BED(\uparrow)$ |
| ✓ | | | 17.99 | 42.29 | 14.45 | 10.14 | 1.57 | 10.05 | 2.14% | 32.71% | 0.4758 |
| ✓ | ✓ | ✓ | 15.39 | **42.17** | 14.57 | 10.17 | 1.46 | 10.24 | 1.59% | **39.99**% | 0.4832 |
| ✓ | ✓ | | **14.96** | 65.97 | **14.75** | **11.37** | **1.16** | **10.40** | **1.57**% | 36.78% | **0.4899** |
| | GT | | 3.61 | 0.82 | 17.12 | 7.61 | 1.35 | 10.81 | 1.07% | 41.95% | 0.5094 |

training is conducted on these two datasets. **Fine-tuning** refers to pre-training on the InterDance dataset followed by fine-tuning on the DD100 dataset.

When trained solely on **DD100**, our method demonstrates high motion quality and interaction quality. Pre-training on InterDance and then **fine-tuning** on DD100 leads to improvements in various metrics compared to training only on DD100, showing that our InterDance dataset provides valuable prior knowledge for generating interactive dances. When trained on **both** InterDance and DD100 datasets, the model achieves optimal performance on most metrics, further demonstrating the high quality and diversity of our dataset.

## H  Instructions for Dancers

During data collection, we provided the dancers with some guidance, as listed below:

(1) **Motion Capture Equipment:** Dancers wear motion capture equipment according to specified procedures and body positions. Dancers don motion capture equipment following specified procedures and body positions.

(2) **Initialization Pose:** Prior to the official performance, dancers must adopt an A pose to initialize the motion capture system.

(3) **Starting Position:** Performances commence from the center of the stage, with movement being both encouraged and permitted.

(4) **Pair Interactions:** Dancers perform in pairs, maximizing interaction during the dance.

(5) **Clarity of Movements:** Movements should be clear and distinct, utilizing highly recognizable actions.

(6) **Harmony with Music:** Careful attention must be paid to synchronizing dance movements with the music beat and ensuring stylistic harmony between the dance and the music.

## I  User Study Screenshots

We conduct our user study through Feishu Docs. The screenshot is shown in Figure 3

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

- In each video, blue represents generated actions, and green represents the input. You do not need to evaluate the green dancer's actions.
- The evaluation is divided into three dimensions:

    i. **Naturalness (Realism) of the Digital Dancer's Movements**:
    - Consider the quality and diversity of the generated actions.
    - Check for issues such as interpenetration, foot sliding, hovering, and sudden displacements.
    - Do not consider the music.

    ii. **Interaction Quality Between Digital Dancers**:
    - Assess the realism and naturalness of the interaction actions.
    - Look for issues such as inaccurate contacts (hovering) and interpenetration at contact points.

    iii. **Coordination Between Digital Dancer's Movements and Background Music**:
    - Consider the consistency between the generated movements and the music style.
    - Check for the rhythm synchronization between music and dance.

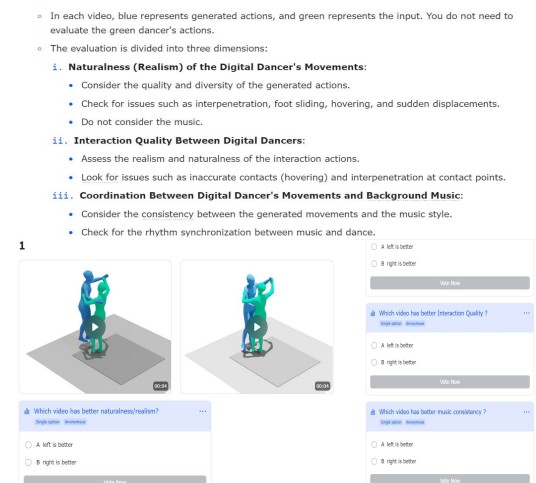

Figure 3: Screenshot of our user study.

Evonne Ng, Hanbyul Joo, Liwen Hu, Hao Li, Trevor Darrell, Angjoo Kanazawa, and Shiry Ginosar. Learning to listen: Modeling non-deterministic dyadic facial motion. In *Proceedings of the IEEE/CVF Conference on Computer Vision and Pattern Recognition*, pp. 20395–20405, 2022.

Li Siyao, Tianpei Gu, Zhitao Yang, Zhengyu Lin, Ziwei Liu, Henghui Ding, Lei Yang, and Chen Change Loy. Duolando: Follower gpt with off-policy reinforcement learning for dance accomapniment. In *ICLR*, 2024.

Jonathan Tseng, Rodrigo Castellon, and Karen Liu. Edge: Editable dance generation from music. In *Proceedings of the IEEE/CVF Conference on Computer Vision and Pattern Recognition*, pp. 448–458, 2023.

