# OpenReview forum: "InterDance: Reactive 3D Dance Generation with Realistic Duet Interactions"
_ICLR.cc/2025/Conference — Submitted to ICLR 2025_

### Official Review · Reviewer_jdiH · 2024-11-02

**Soundness:** 2
**Presentation:** 3
**Contribution:** 2
**Rating:** 5
**Confidence:** 4

**Summary:**

This paper introduce a new large-scale duet dance dataset, which contains 3.93 hours of duet dance mocap data across 15 different music genres. Based on this new dataset, this paper developed an algorithm for dance reaction generation. Specifically, a novel representation on the top of both body vertices and skeleton features is used, and the vanilla diffusion model with classifier guidance is applied for dance motion synthesis. User study, ablation study, and quantitative evaluations are conducted to demonstrate the effectiveness of the proposed framework.

**Strengths:**

**1. The dance dataset is valuable to the community**. This paper proposes a large-scale and high-quality two person dance dataset. Capturing dance motions are expensive and laborious. This will definitely be a nice asset for the community. I appreciate the nice visualization and analysis of the dataset. From table 1, it's well to know this dataset is almost twice the size of the existing largest duet dance dataset, DD100.
**2. Incorporate both body shape and skeleton is a nice touch**. Compared to previous method, this paper propose a new motion representation with both body shape information and skeleton features. Intuitively, this might help on close interactions.
**3. Inference time guidance on penetration and contact is interesting**. The inference time classifier guidance, if being used properly, seems like a nice trial to avoid penetration and contact as a plugin method.
**4. Content is well-organized**. Presentation is clear and easy to follow.
**5. Comprehensive quantitative evaluations.**

**Weaknesses:**

**1. Limited significance and contribution**. This work is not the first one in either duet dance reaction synthesis or duet dance dataset. Though this dataset is larger than the existing one, the overall setting are basically the same and the method are principally a vanilla diffusion model, whose novelty is incremental.
**2. Importance of the task is questionable**. It's not clear why it is important to research on reaction generation, especially for duet dance. In my opinion, the individual dance movements in a duet dance are highly correlated, where each motion alone make little sense. I can not figure out any application that needs to generate the reactive dance movements. It will be more practical to generate movements for both individuals. Though the current method supports generations for the dance of both two individuals, the results looks rather unnatural.
**3. Short dance generation**.  From the results, it seemed the model can not generate quite short dance clips, e.g., 4s. Could the model be scaled to longer sequence?
**4. Insufficient qualitative results**. There are only very few examples for generated results. There are no results for ablation analysis either. It's hard to confirm if the proposed components works as claimed.

**Questions:**

Please refer to the weakness sections.

**Significance**:  the authors need to justify the importance and potential applications of reactive dance generation. Specifically, elaborate on potential use cases or scenarios where generating reactive dance movements could be valuable.

**Short length**:  could provide results or analysis on the model's performance across different sequence lengths, from very short (e.g. 4s) to much longer durations. This would give a clearer picture of the model's scalability.

**Insufficient results**: More diverse examples of generated results across different dance styles and interaction types
Visual comparisons for the ablation studies to illustrate the impact of different components
Failure cases or limitations of the current approach.

---

> ### Author Response · Authors · 2024-11-24
> **Authors‘ Response to Reviewer jdiH**
>
> Thank you for taking the time to review our work and for your valuable feedback. We greatly appreciate your words of “dance dataset is valuable, a nice touch,  interesting, etc”. We hope the following clarifications help to address your concerns.
>
> **W&Q1 Significance and contribution**
>
> Duet dance generation is a challenging task, not only due to the high cost of acquiring high-quality duet datasets but also because current methods do not adequately consider the motion representation and the strategies for generate strong interaction. Our work, on one hand, provides a high-quality duet dataset, and the ablation study in Table 3 of the appendix demonstrates that the inclusion of InterDance effectively alleviates the issue of limited duet dance data. On the other hand, we are the first to incorporate body surface information in duet motion generation tasks and propose a reasonable representation along with a duet motion generation network that supports both reactive generation and simultaneous duet motion generation.
>
> **W&Q2 Importance of the task**
>
> Duet dance is widely used in dance education. However, directly asking two beginners to practice duet dance can be challenging, as it typically requires an experienced dance partner to assist with the teaching, significantly increasing the teaching costs. Therefore, an important application is to generate the follower's movements in a VR environment through human motion generation, helping dancers learn different duet dances.
>
> Meanwhile,  our method can also directly generate duet dances. To the best of our knowledge, we are the first to use a network-based approach for generating music-driven duet dances, and we have achieved promising results. This approach can be widely applied in dance animation and film production.
>
> **W&Q3 Long Dance Generation**
>
> Our work focuses primarily on addressing the fine-grained interaction quality issue. Our method also supports autoregressive generation of long dances using a sliding window approach. Please refer to our project page for videos showcasing long dance generation.
>
> **W&Q4 Qualitative results**
>
> Thank you for your suggestion. We have added more visual results to our project page([https://inter-dance.github.io/](https://inter-dance.github.io/)).

---

> > ### Author Response · Authors · 2024-11-27
> >
> > Dear Reviewer jdiH,
> >
> > We sincerely appreciate the time and effort you have dedicated to reviewing our paper. Your insightful comments and valuable feedback have been extremely beneficial.
> >
> > In response, we have (1)  elaborated on the significance and contributions of our work, (2) demonstrated long-sequence generation, (3) provided additional qualitative results.
> >
> > Should there be any further points that require clarification or improvement, please know that we are fully committed to address them promptly.
> >
> > Thank you once again for your invaluable contribution to our research.
> >
> > Warm regards,
> >
> > The Authors

---

> ### Comment · Reviewer_jdiH · 2024-11-28
> **Response by Reviewer**
>
> Thanks for the response from the authors.  I acknowledge the new dataset can be a nice asset for the community, and the generation of duet dance is a nice trial.
> However, my concern regarding this work remains as follows:
> 1. This work claimed two contributions, one is on a large-scale dataset, while the other is on better reactive duet dance generation. However, these two contribution are both be claimed in Duolando, which proposed duet dance dataset (117 mins), as well as the first effort for reactive dance generation. I would not say a 235 min dataset makes much difference on the quantity comparing to duolando. Meanwhile, though the authors claimed a better motion representation on contact, I did not say particular visual improvement in close-proximity contacts. The generated results remain surreal. Some visual ablations may be helpful. However, the insignifiance of this work is doubtful.
> 2. Reactive duet dance, the main focus of this paper, still looks a relatively minor problem to me. The motions of close-proximity interactions, especially in duet dance, are highly correlated. They are even temporal codependent. It's quite weird to generate the whole other sequence given one's movement. Though this work seems to address the additional task of both duet dance generation, they are not claimed as main contribution.
>
> Overall, I acknowledge that the data are the more the better for the community, and the nice trial on duet dance generation as a side-product. I would like to rise my rating to borderline reject.

---

> ### Author Response · Authors · 2024-12-01
> **Follow up with reviewer jdiH**
>
> Dear reviewer jdiH,
>
> We sincerely appreciate you for raising the score. Your support and encouragement will greatly motivate and advance our work.
>
> Sorry for the delayed response, as we wanted to address your concerns with experimental data. First, we would like to thank you for recognizing our dataset. We believe that both our dataset and code will significantly contribute to research on interactive human motion.
>
> Second, regarding the improvement of our method, as shown in Table 3 of our manuscript and the demo video on our website, we observe that our method outperforms previous SOTA methods on most metrics. We agree with your point that there is still much room for improvement, especially in terms of the close interactions. However, our method has already made an important step forward in interactive motion generation, particularly by considering body surface information for the first time, which will provide a solid foundation for future research. To further demonstrate the effectiveness of our method, we further trained and tested our method on the DD100 dataset to provide an additional comparison with Duolando. The results are as follows, on the DD100 benchmark, our method also outperforms Duolando on most metrics.
>
> | Method | $\text{FID}_k \downarrow$ | $\text{FID}_g  \downarrow$ | $\text{DIV}_k  \uparrow $ | $\text{DIV}_g \uparrow$ | $\text{FID}_{cd} \downarrow$ | $\text{DIV}_{cd} \uparrow$ | $\text{CF}\uparrow$| $\text{BED} \uparrow$ | $\text{BAS}\uparrow$ |
> | --- | --- | --- | --- | --- | --- | --- | --- | --- | --- |
> | Duolando| 43.33| **35.30** | **10.24** | 7.04 | 5.34 | 11.10 | 57.67\%  | 0.2962 | 0.2116 |
> | Ours       | **41.67**| 46.27 | 9.93 | **12.75** | **5.19** | **12.41**   |**62.24\%**   | **0.4208** | **0.2136** |
>
> Third, we agree with your point that generating duet motions simultaneously is an important issue. In this work, we have also focused on this problem, as shown in Table 7 of the manuscript and in the supplementary materials under MORE DETAILS OF DUET DANCE GENERATION and MORE DETAILS OF THE INTERACTIVE DECODER BLOCK.
>
> To our knowledge, we are the first to use a single framework to address both reactive motion generation and duet motion generation. We have provided the community with a new approach that unifies these two tasks. As you mentioned, our work places more emphasis on reactive dance generation because duet interactive motion generation is still in the early stages of research, and there is still significant room for optimization in some strong interactive motions. We first focus on improving the quality of interactive motions generated by our method through reactive dance generation. Thanks to our unified framework, both tasks can benefit. Additionally, in VR scenarios, there are also some applications where the leader's dance of human is used to generate the follower's dance of virtual character, such as for duet dance practice or entertainment purposes.
>
> We hope that these additional experiments have further addressed your concerns. If there are any other questions or suggestions, please do not hesitate to let us know.
>
> Thank you once again for your invaluable contribution to our research.
>
> Warm Regards,
>
> The Authors

---

### Official Review · Reviewer_5vWh · 2024-11-02

**Soundness:** 3
**Presentation:** 3
**Contribution:** 2
**Rating:** 5
**Confidence:** 4

**Summary:**

The submission aims to tackle the generation of duet dance motions, where the modeling of two-person interactions is challenging. To this end, the authors introduce a duet dance dataset featuring body and finger movements with high physical realism, comprising 3.93 hours of music-paired duet dance across 15 different dance genres. In addition, the authors also propose a diffusion-based reactive dance motion generation model with a dedicated module - Interactive Refine Guidance - to enhance the accuracy and realism of interactive dance movements.

**Strengths:**

As always, I admire the contribution made by the authors on the data collection. The duet dance dataset could contribute to the motion generation field, facilitating more future works.

Overall, the paper is well-structured, well-written, and easy to follow.

The submission contains a sufficient number of quantitative evaluations.

**Weaknesses:**

The generated results are not convincing, especially in terms of the interaction between the two performers. See the first on the left of Figure 7, the final generated result (after) has a wrongly placed arm of the blue character, which should presumably hook around the back of the waist of the other performer. And several other results also show incorrect hand interactions. And from the supplementary video results I got an impression that the results of the proposed method do not have significant improvements over those of Duolando. A similar conclusion could also be derived from Table 3, Duolando has similar numbers with Ours, some even better.

The submission lack core technical contributions. While the motion representation described in Section 4.1 makes sense to me, it is not novel. Actually I was wondering whether it is true that the leader's motion representation contains the contact label or not. If yes, requiring contact information from the leader performer could be problematic when applying the trained model in a real-world scenario. Moreover, the Interaction Refine Guidance is not of significant technical contribution either.

Lack of sufficient qualitative results. I would be better to provide more qualitative results, including more comparison results, ablation study results, etc. The numerical numbers sometimes could be inconsistent with the visual results.

**Questions:**

In Section 5, the description of metrics calculation is not rigorous. Is the FID computed by measure the distance between one generated motion and its associated GT motion? Following that, how come the FID computed from the GT set is not 0 - correct me if I am missing anything.

Is it true that the motion representation of the leader includes the contact label?

---

> ### Author Response · Authors · 2024-11-24
> **Authors‘ Response to Reviewer 5vWh**
>
> We sincerely thank you for your time and efforts in reviewing our paper, and your valuable feekback.
> We appreciate your recognition of our dataset, paper, quantitative evaluations and the positive feedback on the motion representation being "makes sense".  We hope the following clarifications help to address your concerns.
>
> **W1 Overall performance.**
>
> See the first on the left of Figure 7, the image here demonstrates that after applying the interaction guidance we proposed, we effectively resolved the self-penetration issue, and this movement is physically plausible. Duet dance generation is a challenging task. Although there are still some shortcomings in performance, our method represents a significant improvement over previous work, especially with the inclusion of body surface modeling.
>
> **W2 Comparasion with Duolando**
>
> In our demo video, our method demonstrates reasonable interaction quality, while Duolando sometimes fails to track the leader's motion effectively, as seen between 2:20 and 2:50.
>
> In our comparison experiments, we achieve state-of-the-art results in most metrics for both reactive dance generation (Table 3) and duet dance generation (Table 7), with our method significantly outperforming all previous methods, including Duolando, especially in terms of interaction quality.
>
> **W3 Technical contributions.**
>
> Regarding motion representation, our proposed approach is not a simple combination of existing methods. Unlike HumanML3D[1], which uses velocity to express root translation and rotation but lacks surface vertices, and COINS[2], which does not account for canonical space processing, our representation introduces a novel approach specifically designed for interactive motion generation.
>
> Regarding the method, our framework uniquely supports both reactive generation and duet dance generation within a single structure. It effectively leverages contact guidance and penetration guidance to optimize fine-grained interactions, marking the first time these techniques have been introduced in duet motion generation research.
>
> **W4 Qualitative results**
>
> Thank you for your suggestion. We have added more visual results to our project page([https://inter-dance.github.io/](https://inter-dance.github.io/)).
>
> **Q1 FID calculation**
>
> We calculate the Fréchet Inception Distance (FID) between the generated motion and the real data across the entire InterDance dataset following previous works [3,4]; therefore, the FID of the GT test set is not zero. We use motion kinematic features [5] and graphical features [6] to compute the FID distance.
>
> Due to the complexity of dance movements and the scarcity of dance data, directly comparing the FID difference between the generated motions and the dance data test set is unlikely to provide an effective evaluation. Therefore, comparing the FID between generated motions and the entire dataset is a widely used approach. A large number of dance generation works also follow this FID calculation standard, as it provides a more accurate and objective evaluation of dance generation algorithms.
>
> **Q2 Contact label**
>
> Our motion representation includes contact labels, and we leverage contact guidance in our method to further utilize this information for promoting more accurate and fine-grained interactions. In duet dance generation, the motion features of both dancers, including the contact labels, are predicted. In reactive dance generation, the the leader’s motion is provided, which also include contact label. This setup is motivated by the following considerations:
>
> 1. In some practical applications, even given a leader motion without  contact label, we can retrieval the nearest neighbour contact labels from the training set, making our method suitable for more practical applications.
> 2. By replacing the contact labels in the leader’s motion, we can do data enhancement to get more diverse and high-quality interactive motions. Obtaining paired interactive motions is inherently challenging, and this approach allows us to overcome this difficulty while advancing the research on duet motion generation. We will explore this issue further in our future work.
> 3. A unified representation and algorithmic framework can be applied to both reactive dance generation and duet dance generation.
>
> [1] Generating Diverse and Natural 3D Human Motions From Text.
>
> [2] COINS: Compositional Human-Scene Interaction Synthesis with Semantic Control.
>
> [3] Bailando: 3d dance generation by actor-critic gpt with choreographic memory.
>
> [4] Duolando: Follower gpt with off-policy reinforcement learning for dance accompaniment.
>
> [5] Fmdistance: A fast and effective distance function for motion capture data.
>
> [6] Efficient content-based retrieval of motion capture data.

---

> > ### Author Response · Authors · 2024-11-27
> >
> > Dear Reviewer 5vWh,
> >
> > We sincerely appreciate the time and effort you have dedicated to reviewing our paper. Your insightful comments and valuable feedback have been extremely beneficial.
> >
> > To provide a more comprehensive and accurate response to your comments, we have further edited our previous response. In summary: we have: (1) clarified the technical contributions of our motion representation and method, (2) provided additional visualization videos, and (3) answer questions regarding FID and contact labels to resolve your concerns.
> >
> > Should there be any further points that require clarification or improvement, please know that we are fully committed to address them promptly. Your further comments and insights would be invaluable to us as we strive to enhance our work.
> >
> > Thank you once again for your invaluable contribution to our research.
> >
> > Warm regards,
> >
> > The Authors

---

> > ### Comment · Reviewer_5vWh · 2024-11-27
> > **Reviewer Response**
> >
> > Thanks for your reply.  It seems like the contact label is important in the proposed framework.
> >
> > While the authors highlight some upsides we can have with the contact label input, they do not indeed outweigh the concerns of requiring the contact label as input. Given a fact that the acquisition of the contact label in the downstream applications is itself an open problem - likely not a easy one with a perfect solution, I was wondering what the result looks like if we ablate the contact label in the current framework.
> >
> > Additionally, considering the marginal superiority of the proposed method over Duolando, I am also interested in seeing a comparison between the contact label-ablated variant of the proposed method and Duolando.

---

> ### Author Response · Authors · 2024-11-28
>
> Dear Reviewer 5vWh,
>
> Thank you for reviewing our response and engaging in a positive discussion. We are currently conducting additional experiments based on your suggestions and striving to complete them as quickly as possible. Thanks for your patience.
>
> Once again, we sincerely appreciate your invaluable contribution to our research.
>
> Warm regards,
>
> The Authors

---

> ### Author Response · Authors · 2024-12-01
> **Follow up with reviewer 5vWh**
>
> Dear reviewer 5vWh,
>
> Tanks for your further valuable suggestions.
>
> We followed your suggestions and conducted experiments by removing the contact label from the given leader. In the table below:
>
> 1. The first row  **Ours (Repaint w/o leader label)**, indicates that we applied the diffusion repainting technique during the inference phase to generate the follower dance under the control of leader motion (without the leader contact label).
>
> 2. The second row, **Ours (Trained w/o leader label)**, shows the results when we retrained our network without providing the leader contact label to generate the follower dance, and then use the new trained model to generate follower dance conditioned by the given music and leader dance (without the leader contact label).
> **The videos results are also updated to our project page** (https://inter-dance.github.io/).
>
> 3. The third row represents the metrics of our method from the manuscript, where the given leader motion includes the leader contact label.
>
> 4. The last row shows the metrics of Duolando, trained and tested on the InterDance dataset, also without given contact label. Ours (Train w/o leader label) outperforms Duolando in metrics such as FID, PR, CF, and others. It even achieves the highest metrics in BED and BAS.
>
>
> | Method | $\text{FID}_k \downarrow$ | $\text{FID}_g \downarrow$ | $\text{DIV}_k \uparrow$ | $\text{DIV}_g \uparrow$ | $\text{FID}_{cd} \downarrow$ | $\text{DIV}_{cd} \uparrow$ | $\text{PR} \downarrow$ | $\text{CF} \uparrow$ | $\text{BED} \uparrow$ | $\text{BAS} \uparrow$ |
> | --- | --- | --- | --- | --- | --- | --- | --- | --- | --- | --- |
> | Ours (Repaint w/o leader label) | 75.08 | 8.64 | **10.95** | 5.92 | 2.59 | 10.34 | **0.08%** | 4.78% | 0.3531 | 0.2002 |
> | Ours (Train w/o leader label)     | 68.63 | 8.07 |   10.37     | 6.04 | 1.93 | 10.86 | 0.28% | 5.76% | **0.3753** | **0.2013** |
> | Ours                               | **65.96** | **8.02** |   10.53     | **6.07** | **1.51** | **10.89** | 0.36% | **6.99%** | 0.3644 | 0.1992 |
> | Duolando | 89.79 | 8.14 |   10.13     | **6.07** | 4.19 | 9.78 | 0.19% | 3.49% | 0.2608 | 0.1995 |
>
> Additionally, we trained and tested our method on the DD100 dataset to provide a more fair apple-to-apple comparison with Duolando, both methods generate follower dance conditioned on the given music and leader dance without contact label. The results are as follows, on the DD100 benchmark, our method also outperforms Duolando on most metrics.
>
> | Method | $\text{FID}_k \downarrow$ | $\text{FID}_g  \downarrow$ | $\text{DIV}_k  \uparrow $ | $\text{DIV}_g \uparrow$ | $\text{FID}_{cd} \downarrow$ | $\text{DIV}_{cd} \uparrow$ | $\text{CF}\uparrow$| $\text{BED} \uparrow$ | $\text{BAS}\uparrow$ |
> | --- | --- | --- | --- | --- | --- | --- | --- | --- | --- |
> | Duolando| 43.33| **35.30** | **10.24** | 7.04 | 5.34 | 11.10 | 57.67\%  | 0.2962 | 0.2116 |
> | Ours       | **41.67**| 46.27 | 9.93 | **12.75** | **5.19** | **12.41**   |**62.24\%**   | **0.4208** | **0.2136** |
>
> We hope that these additional experiments have further addressed your concerns. If there are any other questions or suggestions, please do not hesitate to let us know.
>
> Thank you once again for your invaluable contribution to our research.
>
> Warm Regards,
>
> The Authors

---

### Official Review · Reviewer_UHsx · 2024-11-03

**Soundness:** 3
**Presentation:** 3
**Contribution:** 3
**Rating:** 6
**Confidence:** 3

**Summary:**

The paper study 3D dance generation. It introduces a large-scale duet dance dataset. To better model interactive motion, it also presents a motion representation method and an enhanced diffusion-based framework featuring an interaction-refined guidance sampling strategy.

**Strengths:**

### Originality
The originality of the paper are from three aspects:
1. a new large-scale dataset;
2. a new motion representation;
3. an interaction refine guidance sampling strategy.

### Quality
The generation results are good, especially visual outcomes. And the experiments are comprehensive, which can well validate the paper's claim. Furthermore, the code and project page will be released to enhance the reproducibility.

### Clarity
The paper is well-organized, and all the concepts are explained clearly. Furthermore, the technical details are also presented clearly.

### Significance
The proposed dataset is good for duet dance generation, and the introduced motion representation and sampling strategy are essential for more effectively modeling interactive motions.

**Weaknesses:**

The reviewer does have some concerns as listed below:

1. There is a lack of comparison with state-of-the-art (SOTA) methods on the DD100 dataset.

2. According to the ablation study, it appears that $L_{RO}$ and Vertices may hinder motion quality.

3. I disagree with the paper's assertion that "more interaction brings more penetration." I believe a well-designed model should enhance interaction without a corresponding increase in the penetration rate. Therefore, it would be better that the paper develops a new metric that integrates both penetration rate (PR) and contact frequency (CF) to better assess the model's performance.

4. The novelty of the method warrants concern. Regarding the dataset, while the proposed dataset is twice the size of the existing DD100 dataset [1], I don't believe it offers substantial improvements for this task. As for the method, the proposed motion representation seems to be a trivial combination of two existing representations [2, 3]. For the sampling guidance strategy, I recommend that the paper cite related works, as using gradients to guide the denoising process is a common practice in diffusion-based generation. In conclusion, it would be better that the paper pays more attention to clearly articulating the novelty and uniqueness of the proposed dataset and methods.

These weaknesses may diminish the overall contribution of this paper.

**Questions:**

Please respond to the concerns above.

---

> ### Author Response · Authors · 2024-11-24
> **Authors‘ Response to Reviewer UHsx**
>
> We sincerely thank you for your time and efforts in reviewing our paper, and your valuable feekback.*  We greatly appreciate your recognition of  a new large-scale dataset,  a new motion representation and an interaction refine guidance sampling strategy. Please see below for our responses to your comments.
>
> **W1: Comparison on DD100 dataset**
>
> Thanks for pointing this. While we conducted comprehensive comparisons with various SOTA methods on the InterDance dataset, including reactive and duet dance generation, we did not include this comparisons due to we focus on our dataset and method.
>
> **W2: The effectivess of  $\mathcal{L}_{RO}$ and vertices**
>
> The $\mathcal{L}_{RO}$ and vertices are highly effective, particularly in optimizing strong interactive motions.
> As shown in Table 3, **Motion Quality** primarily evaluates the quality of individual motions, while **Interaction Quality** assesses the quality of interactions.
> Although adding them results in a deterioration in FID}_k, the metric of **Interaction Quality**, such as FID_cd , shows a significant improvement.
>
> **W3: The relationship between interaction and penetration**
>
> We agree with your comments and have made the corresponding modifications. Please refer to our manuscript, where the revised sections are highlighted in blue.
>
> **W4: The novelty**
>
> Regarding the dataset, we conducted ablation studies as shown in Table 3 of the appendix. Compared to training solely on the DD100 dataset, joint training on both DD100 and InterDance significantly improves various metrics. Additionally, our dataset features a broader range of dance styles, making it a substantial contribution to the fields of duet dance generation and interactive motion synthesis.
>
> For motion representation, our proposed approach is not a simple combination of existing methods. Unlike HumanML3D, which uses velocity to express root translation and rotation but lacks surface vertices, and COINS, which does not account for canonical space processing, our representation introduces a novel approach specifically designed for interactive motion generation.
>
> Regarding the method, our framework uniquely supports both reactive generation and duet dance generation within a single structure. It effectively leverages contact guidance and penetration guidance to optimize fine-grained interactions, marking the first time these techniques have been introduced in duet motion generation research.
>
> Finally, we have also followed your kind suggestions and added references to strengthen our work.

---

> > ### Author Response · Authors · 2024-11-27
> >
> > Dear Reviewer UHsx,
> >
> > We sincerely appreciate the time and effort you have dedicated to reviewing our paper. Your insightful comments and valuable feedback have been extremely beneficial.
> >
> > In response, we have (1) clarified the effectivess of $\mathcal{L}_{RO}$ and vertices, (2) corrected the descriptions regarding penetration and interaction, and (3) elaborated on our contributions and novelty to further address your concerns.
> >
> > Should there be any further points that require clarification or improvement, please know that we are fully committed to address them promptly.
> >
> > Thank you once again for your invaluable contribution to our research.
> >
> > Warm regards,
> >
> > The Authors

---

> > > ### Comment · Reviewer_UHsx · 2024-11-27
> > >
> > > Thank you for providing the qualitative scores for the DD100 (ICLR24) dataset in Appendix-Table 5. However, I noticed that the reported scores for 'GT' appear inconsistent with those presented in the original DD100 paper. Could the authors elaborate on this discrepancy?

---

> > > > ### Author Response · Authors · 2024-11-27
> > > >
> > > > Dear Reviewer UHsx,
> > > >
> > > > Thank you reviewing our response and possitve discussion. Although both DD100 and InterDance are in the SMPL-X format, their coordinate systems differ. DD100 adopts the Blender coordinate system (z-axis up), which facilitates easier rendering and visualization, while InterDance uses the SMPL coordinate system (y-axis up) for better compatibility with other SMPL-based datasets commonly used in the academic community.
> > > >
> > > > To conduct the ablation experiments presented in Appendix Table 5 and enable our method to train on both datasets simultaneously, we converted the motion data of DD100 into the SMPL coordinate system, which is consistent with InterDance. After this conversion, the ground truth metrics of DD100 also changed accordingly. Therefore, we remeasured the metrics on the test set of DD100.
> > > >
> > > > Thanks for your comments, we have added the corresponding descriptions to the manuscript in response to your comments.
> > > >
> > > > Warm regards,
> > > >
> > > > The Authors

---

> > > > > ### Comment · Reviewer_UHsx · 2024-11-28
> > > > >
> > > > > Thank you for your response.
> > > > >
> > > > > I still believe it is important to compare with other approaches on DD100, following its settings for consistent benchmarking. The main reason is that the paper follows DD100, and the contributed dataset is only twice the size of DD100, which may not lead to significant improvements for this task.
> > > > >
> > > > > As the authors mentioned, the difference in inconsistency scores lies in the coordinate system. Does this mean we can simply recompute the scores by converting the coordinate system of the obtained results without needing to retrain the model? I believe the requested "DD100 comparison" study is straightforward to conduct and could help readers better understand the two datasets and the proposed approach.
> > > > >
> > > > > Overall, I will maintain my borderline accept rating. My positive rating is primarily due to the contributed dataset. While its scale remains relatively small, it can serve as a useful supplement to DD100. To me, the technical contribution is incremental and requires further evaluation.

---

> ### Author Response · Authors · 2024-12-01
> **Follow up with reviewer UHsx**
>
> Dear reviewer UHsx,
>
> We appreciate your responses and further valuable suggestions.
>
> Apologies for the delayed response, as we have struggle to conducting experiments over the past few days. We have finally tested our method on the DD100 benchmark. As shown in the table below, we transformed the DD100 data into our motion representation and coordinate system for training, and then converted the generated results back to the original motion space of DD100 for testing. Compared to the metrics listed in the original Duolando paper, our method outperforms Duolando on most metrics. This is because our method’s motion representation incorporates human surface information, and the diffusion structure makes it easier to optimize fine-grained interaction quality.
>
> | Method | $\text{FID}_k \downarrow$ | $\text{FID}_g  \downarrow$ | $\text{DIV}_k  \uparrow $ | $\text{DIV}_g \uparrow$ | $\text{FID}_{cd} \downarrow$ | $\text{DIV}_{cd} \uparrow$ | $\text{CF}\uparrow$| $\text{BED} \uparrow$ | $\text{BAS}\uparrow$ |
> | --- | --- | --- | --- | --- | --- | --- | --- | --- | --- |
> | Duolando| 43.33| **35.30** | **10.24** | 7.04 | 5.34 | 11.10 | 57.67  | 0.2962 | 0.2116 |
> | Ours       | **41.67**| 46.27 | 9.93 | **12.75** | **5.19** | **12.41**   |**62.24**   | **0.4208** | **0.2136** |
>
> Regarding to the technical contribution, we are the first to use a single framework to address both reactive motion generation and duet motion generation. We have provided the community with a new approach and motion representation that unifies these two tasks. We believe that our dataset and code will significantly contribute to research on interactive human motion.
>
> We hope that these additional experiments have further addressed your concerns. If you have any other questions or suggestions, please do not hesitate to let us know.
>
> Thank you once again for your invaluable contribution to our research.
>
> Warm Regards,
>
> The Authors

---

### Official Review · Reviewer_yXpV · 2024-11-03

**Soundness:** 3
**Presentation:** 3
**Contribution:** 2
**Rating:** 6
**Confidence:** 4

**Summary:**

The paper introduces a new duet dance dataset that includes a variety of dance genres and features high-quality motion capture (MoCap) data. This dataset is larger in scale compared to previous ones. The authors also propose a new motion representation designed to better capture interactive movements, implemented within a novel diffusion-based framework. Experimental results demonstrate the effectiveness of both the new dataset and the proposed algorithm.

**Strengths:**

The overall paper is well written and structured.
1. The new dataset captures detailed body and finger movements, which are valuable for the motion generation community.
2. The proposed motion representation includes multiple components: positional space, joint data, hand movements, surface information, and canonical space. This combination aims to make generated motions more realistic.
3. The authors propose a diffusion-based model for reactive dance motion generation, incorporating an Interactive Refine Guidance mechanism to improve the accuracy and realism of interactive dance movements.

**Weaknesses:**

However, there are several areas for improvement:

1. Since the dataset is a major contribution of this paper. The dataset description should include not only the total hours of data but also detailed information on the number of frames and sequences for each category.

2. For the 655 vertices used in the study, how were they selected from SMPL-X? Were they sampled randomly or in a way that preserves the topology?

3. $L_f$ for foot contact loss is not clearly stated.

4. The symbol $f$ is used in two different contexts in the paper: it denotes foot contact in Equation (1) and forward motion in Equations (2), (3), and (4).

5. One important realted work [1] is missing.
[1] Listen, denoise, action! Audio-driven motion synthesis with diffusion models. SIGGRAPH 2023.

6. Line 72     priors, without -> prior without

**Questions:**

For the dataset, 1) Compared to text-to-motion datasets, does each category in this dataset feature multiple pairs of dancers, or just one pair per category? 2) Since human participants were involved in data collection and the data will be released in the future, did all dancers sign consent forms?

**Details Of Ethics Concerns:**

Since the dataset is a major contribution of this paper, and the authors plan to release the captured data, did all participants sign consent forms?

---

> ### Author Response · Authors · 2024-11-24
> **Authors‘ Response to Reviewer yXpV**
>
> Thanks for your positive recommendation and valuable suggestions. We are happy to see your recognition of our proposed dataset, motion representation and method. Please see below for our responses to your comments.
>
> **W1: More details of dataset**
>
> We  summarize the dance data for different genres in terms of total duration (in minutes), number of frames, number of sequences, and their percentage of the entire InterDance dataset. We have added this table to our manuscript.
>
> | Genre | Minutes | Frames | Sequences | Percentage |
> | --- | --- | --- | --- | --- |
> | Jive | 8.20 | 59006 | 3 | 3.48% |
> | Chacha | 11.25 | 81000 | 4 |4.78% |
> | HanTang | 15.44 | 111154 | 6 |6.56% |
> | Kpop | 53.79 | 387262 | 21 |22.84% |
> | Jazz | 6.61 | 47,610 | 4 |2.81% |
> | Samba | 10.69 | 76982 | 6 |4.54% |
> | Waltz | 10.79 | 77,674 | 8 |4.58% |
> | Rumba | 27.11 | 195162 | 7 |11.51% |
> | Dai | 11.95 | 86034 | 6 |5.07% |
> | HipHop | 31.87 | 229432 | 10 |13.53% |
> | Wei | 11.78 | 84822 | 6 |5.00% |
> | ShenYun | 18.44 | 132758 | 8 |7.83% |
> | DunHuang | 6.41 | 46172 | 3 |2.72% |
> | Urban | 6.58 | 47390 | 4 |2.80% |
> | Miao | 4.58 | 32,978 | 3 |1.95% |
>
> **W2: How to select the 655 vertices?**
>
> We follow the settings of COINS[1] and PyMAF-X[2] to downsample the vertices of SMPL-X. These 655 vertices are not randomly sampled and preserve the topology, allowing for direct rendering of the human mesh on these 655 vertices. We have added these details in our manuscript.
>
> **W3: Foot contact loss is not clearly stated**
>
> Thanks for your suggestion, we have add the formulation to our manuscript. Please see Line 318-319, the revised texts in our manuscript are marked in blue.
>
>
> **W4: The symbol $f$ used for foot contact and follower motion**
>
> Thanks for your thorough review. We have marked the symbols related to foot contact as $foot$ and made the revisions in our manuscript.
>
> **W5: One important realted work**
>
> Thanks for your reminder. This is indeed an important work. We have added a citation for it and included a comparison with Motorica Dance in Table 1 of our manuscript.
>
> **W6: Line 72 priors, without -> prior without**
> Thanks for your detailed review. We have fixed this in our manuscript.
>
> **Questions of dancers**
>
> Each category has multiple pairs of dancers. All the dancers have signed consent forms, allowing us to publicly release the data.
>
> **References**
>
> [1] Compositional humanscene interaction synthesis with semantic control
>
> [2] PyMAF-X: Towards Well-aligned Full-body Model Regression from Monocular Images

---

> > ### Author Response · Authors · 2024-11-27
> >
> > Dear Reviewer yXpV,
> >
> > We sincerely appreciate the time and effort you have dedicated to reviewing our paper. Your insightful questions and valuable feedback have been extremely beneficial.
> >
> > In response, we have (1) provided more details about our dataset, (2) explained the method for selecting the 655 vertices and added the foot contact loss formulation in Line 318 of the manuscript, and (3) corrected and further reviewed the symbols, sentences and references following your detailed comments.
> >
> > Should there be any further points that require clarification or improvement, please know that we are fully committed to address them promptly.
> >
> > Thank you once again for your invaluable contribution to our research.
> >
> > Warm Regards,
> >
> > The Authors

---

> ### Author Response · Authors · 2024-12-01
> **Follow up with reviewer yXpV**
>
> Dear reviewer yXpV,
>
> We wish to express our sincere appreciation to you for recognizing the substantial significance of our InterDance's contribution.
>
> Your acknowledgment holds great importance to us and serves as a meaningful validation of our dedicated efforts to advance this research.
>
> If you have any other questions or suggestions, please do not hesitate to let us know.
>
> Thank you once again for your invaluable contribution to our research.
>
> Warm Regards,
>
> The authors

---

### Official Review · Reviewer_oJee · 2024-11-04

**Soundness:** 2
**Presentation:** 3
**Contribution:** 3
**Rating:** 6
**Confidence:** 5

**Summary:**

This paper presents a mocap dataset for couple dances, a novel motion representation that takes contact into account, and a diffusion-based model for predicting conditional motion during couple dance.

**Strengths:**

* The proposed mocap dataset is a good contribution to the community especially as it contains enough fine-grained information to capture contact points between people.
* The proposed motion representation prioritizes correctly predicting contact, something that is often missing from similar works.
* The proposed diffusion-based method seems to achieve competitive results (though see weaknesses and questions for issues with the evaluation methodology).

**Weaknesses:**

* Evaluation:
    * Simple strong baselines missing from the quantitative and qualitative experiments:
        * Return a NN motion clip of a follower from the training set where the distance is calculated based on motion-based distance
        * Return a NN motion clip of a follower from the training set where the distance is calculated based on a SOTA music embedding distance
        * Mirror the motion of the leader.
    * Metrics:
        * Missing a measure of correlation between the dancers in comparison to ground truth (see questions below)
    * Baselines:
        * Missing an apples-to-Apples comparison with existing baselines on the datasets on which they were trained:
            * Result tables in main and appendix only seem to test Duolando on the InterDance dataset or train the proposed method on DD100 (Table 3 in the appendix E) but I couldn’t find an apples-to-apples comparison of the proposed method trained and tested on the existing DD100 to Duolando trained and tested on DD100 (allowing Duolando to be tested on the dataset on which it was trained.
            * This apples-to-apples comparison is crucial in the case of Duolando which learns a motion codebook from training data and therefore is not suited to be tested on out-of-distribution dance genres.
* Exposition:
    * Missing some implementation details for the experimental setup (see details below in the questions section)
    * Missing details of the user study: where were participants recruited from, how were they compensated, in what format were videos presented, in what resolution, was music included and did the experimenters verify it was in use, etc.
* Smaller notes:
    * The caption of Table 1 is not entirely factual as stated since InterHuman has Strong interaction and a longer total duration: “Among duet dance datasets with strong interaction, InterDance features the widest range of 15 dance genres, the longest average duration per sample at 142.7 seconds, and the longest total duration of 3.93 hours.”
* Limitations:
    * Limitations are not properly discussed. As in all studies, there are limitations in both dataset and method. These are to be expected, but should also be mentioned.
        * The dataset is indeed larger than existing mocap ones, but is still a small dataset when compared to methods that can rely on image-based 3D lifting. There is a balance here between data quality and quantity, but that should be mentioned.
        * There are limitations to using diffusion-based generative models for motion prediction in comparison with autoregressive methods, namely the full trajectory is predicted at once, without training to predict longer sequences in a sliding-window fashion.
        * This is the only limitation mentioned: “The potential societal impact is that, as dance generation and human interaction technology become more advanced, highly realistic virtual humans might lead users to become so immersed in the virtual world that they detach from real-world participation.” (L 531). However, if this technology becomes more advanced, more imminent limitations may present themselves. Namely, music-to-dance and person-to-person prediction of dance may be used by AI systems to replace human dancers and choreographers, leading to a less creative human kind and numerous copyright issues.

**Questions:**

* Dataset: why not combine with DD100 to create a larger total dataset?
* Motion representation: if the root translation and angle is modeled jointly with the body pose (joints and vertices) it seems to me that translation, orientation, and pose would be coupled together. i.e. a person lifting their leg at point A in space would need a different representation then a person lifting their leg in the same way at point B in space. This seems to lead to needing more data in order to learn motion priors from the data as opposed to representations that would decouple pose from location and orientation. I am not listing this as a limitation yet as I would like to hear your thoughts about this issue. Would you be so kind as to explain this to me?
* Evaluation:
    * Metrics:
        * How is FID calculated?
        * While cross distance captures the distance between the two dancers, I am missing some measure of the correlation between the motion dynamics of the two dancers (see, for example paired FD here: https://arxiv.org/pdf/2204.08451). Do you think that cross distance captures this already somehow? If so, please explain. If not, I would suggest adding a measure of correlation in comparison to the ground truth correlation.
    * Baselines:
        * how are Edge and InterGen modified for benchmarking on this task (as they were designed a trained for different tasks)? Details are not given in the manuscript.
        * Was Duolando trained on the proposed InterDance dataset for the experiments presented? Or was it trained on its original DD100 dataset and tested on InterDance?
        * The following explanation regarding the performance of Duolando is cryptic to me. Please explain what you meant by it (line 421) “Compared with Duolando, which is carefully designed for reactive dance generation, our method performs better in most metrics. This is because Duolando employs a two-stage training framework that stores motions in a codebook, making it difficult to optimize fine-grained interactions.”
* Result videos:
    * Are the result videos on unseen test examples or from the train/val set?
    * Are the other methods in the result videos tested on the newly proposed dataset or on their original datasets on which they were trained? Was testing conducted on the test sets of the other methods in comparison?
    * Why is EDGE not in the results videos?
    * Please provide results also on randomly-chosen (rather than cherry-picked) examples.

---

> ### Author Response · Authors · 2024-11-24
> **Authors‘ Response to Reviewer oJee (Part 1/2)**
>
> Thank you for taking the time to review our work and for your valuable feedback. We greatly appreciate your recognition of our contributions, including the mocap dataset with  fine-grained contact information, the proposed motion representation correctly predicting contact, and the diffusion-based method and achieve competitive results. We hope the following clarifications help to address your concerns.
>
> ## **Response of Weakness:**
>
> **Evaluation 1: Simple strong baselines**
>
> We have followed your suggestions and added these baselines in our manuscript. The table below shows the results, tested on the InterDance dataset.
>
> NN-motion refers to “Return a NN motion clip of a follower from the training set where the distance is calculated based on motion-based distance”.
> Given an input leader motion, we find its nearest neighbor from the training set and use its corresponding follower segment as the prediction.The distance is calculated based on the MSE distance of the motion clips.
>
> NN-music refers  to “Return a NN motion clip of a follower from the training set where the distance is calculated based on a SOTA music embedding distance”.
> Given an input music clip on InterDance test set, we find its nearest neighbor music clip from the training set and use its corresponding follower motion as the prediction. The distance is calculated based on the MSE distance of the music features extracted by the advanced music feature extractor Jukebox[2].
>
> | Method | $\text{FID}_k$ | $\text{FID}_g$ | $\text{DIV}_k$ | $\text{DIV}_g$ | $\text{FID}_{cd}$ | $\text{DIV}_{cd}$ | $\text{PR}$ | $\text{CF}$ | $\text{BED}$ | $\text{BAS}$ |
> | --- | --- | --- | --- | --- | --- | --- | --- | --- | --- | --- |
> | NN-motion | 48.4 | 7.17 | 13.22 | 8.19 | 2.07 | 10.81 | 0.52% | 5.54% | 0.2356 | 0.1904 |
> | NN-music | 24.01 | 4.34 | 16.53 | 6.78 | 14.07 | 12.76 | 0.18% | 3.04% | 0.1966 | 0.1890 |
>
> As table below shows, Ours (mirror) refers  to “Mirror the motion of the leader”. We use the mirrored leader motion and corresponding music from the InterDance test set as conditions to generate the follower motion and test its metrics.
>
> ---
>
> | Method | $\text{FID}_k$ | $\text{FID}_g$ | $\text{DIV}_k$ | $\text{DIV}_g$ | $\text{FID}_{cd}$ | $\text{DIV}_{cd}$ | $\text{PR}$ | $\text{CF}$ | $\text{BED}$ | $\text{BAS}$ |
> | --- | --- | --- | --- | --- | --- | --- | --- | --- | --- | --- |
> | GT (mirror) | 9.84 | 2.17 | 15.62 | 7.41 | 1.41 | 10.93 | 0.55% | 10.01% | 0.3471 | 0.1901 |
> | **Ours (mirror)** | 64.4 | 8.16 | 10.63 | 6.02 | 3.02 | 9.53 | 0.45% | 7.87% | 0.3681 | 0.2017 |
>
> **Evaluation 2:  Metrics**
>
> We used a comprehensive set of metrics to evaluate Interaction Quality, including FID_cd to measure interaction quality, DIV_cd to assess the diversity of interaction motions, PR and CF to evaluate the physical realism of the interaction, and BED to measure the rhythmic alignment of the duet dance.
>
> The PCC in [1] can be used to measure the covaries of two 1D time series data. However, in duet dance, a more appropriate metric is BED.
>
> The TLCC correlation in [1] can analyze the correlation between two sequences at different time lags, making it more suitable for measuring the correlation of two time series with causal relationships, therefore is commonly used in two-person dialogue scenarios. However, many movements in duet dances occur synchronously, involving more complex interaction dynamics. Therefore, measuring the FID distance of the cross-distance can better assess interaction quality by comparing the similarity between the generated motions and the Ground Truth dataset.
>
> **Evaluation 3: Baselines**
>
> To ensure a fair comparison, we trained and tested several SOTA methods, including Duolando, on the InterDance dataset. The experimental results are presented in Table 3 of the manuscript.  Since Duolando was also trained on the InterDance dataset, this constitutes an apples-to-apples comparison.
>
> **Exposition: Details of the user study**
>
> We recruited 40 participants for the user test, including 14 individuals with a dance background, consisting of students with dance experience and professional dancers from dance training schools, as well as 26 randomly recruited participants with no dance background. Each participant received a payment of 50 RMB. None of the participants had prior knowledge of the project before the test. To ensure fairness, we randomly shuffled the order of the results generated by our method and the different comparison results. The videos were presented in 30fps, 1280x1024 mp4 format, accompanied by background music. We have updated the details of the user test in the manuscript.
>
> **Smaller notes: The caption of Table 1**
>
> Thank you for your thorough review. We have corrected this description in the manuscript.
>
> **Limitations**
>
> Thank you for your suggestions. We have added more discussion in the Limitations section.

---

> ### Author Response · Authors · 2024-11-24
> **Authors‘ Response to Reviewer oJee (Part 2/2)**
>
> ## **Response of Questions:**
>
> **Dataset**
>
> Thanks for your suggestion. As shown in Table 3 of the appendix, we have already jointly trained the neural network on DD100 and InterDance. We will provide the code to support the combined use of both datasets.
>
> **Motion representation**
>
> Decoupling root translation and angle information is more conducive to learning motion priors, and this is exactly what we have done. Our representation records the body’s joint and vertex information relative to the root node, excluding global orientation. For root translation and angle, we used absolute position and angle rather than velocity and angular velocity because, in duet interaction
> motions, the relative position and orientation between the two individuals are critically important. Using velocity and angular velocity would make it challenging to optimize their relative position and orientation effectively.
>
> **Evaluation 1: Metric**
>  - We calculate the Fréchet Inception Distance (FID) distance  between the generated motion and the real data in whole InterDance dataset following Duolando. We use motion kinematic feature [3] and graphical feature [4] to calculate the FID distance.
>  - Please see weaknesses response above.
>
> **Evaluation 2: Baselines**
>
> - For EDGE, we encoded the leader motion features using a Transformer and concatenated them with music features as the new condition to generate follower motion. For InterDance, we replaced the text encoder with a Transformer-based music encoder. These changes have been updated in the manuscript. We have added these details to the manuscript.
> - Duolando was trained on the InterDance dataset and evaluated on its test set to ensure a fair comparison.
> - We argue that Duolando’s VQ-VAE+GPT framework encodes motion into a coarse-grained codebook. After training, the weights of the VQ-VAE are frozen, and therefore the following GPT network is challenging to learn fine-grained interactions. Although Duolando incorporates some optimization modules to enhance duet interaction performance, our method demonstrates superior fine-grained interaction quality compared to Duolando.
>
> **Result videos**
>
> - The generated videos we showcase are all produced conditioned on the test set's music or leader dance
> - To ensure a fair visual comparison, we trained other methods on InterDance and tested both our method and the others on the InterDance test set under the same conditions.
> - EDGE is a single-person motion generation framework, and while it demonstrates excellent performance in solo dance generation, it lacks optimization for duet motion generation. As a result, the generated duet dance is noticeably inferior to other methods. We believe this is not comparable, so we did not showcase EDGE's generated results.
> - In our result videos, you can see some generated outputs. We have added more generation results. Please refer to our project page [https://inter-dance.github.io/](https://inter-dance.github.io/) for further details.
>
> **References**
>
> [1] Learning to Listen: Modeling Non-Deterministic Dyadic Facial Motion.
>
> [2] Jukebox: A Generative Model for Music.
>
> [3] Fmdistance: A fast and effective distance function for motion capture data.
>
> [4] Efficient content-based retrieval of motion capture data.

---

> > ### Comment · Reviewer_oJee · 2024-11-25
> > **Revision?**
> >
> > Thank you for your detailed answers. Can you upload your paper revision please?
> >
> > Much appreciated

---

> > > ### Author Response · Authors · 2024-11-25
> > > **Response of Revision**
> > >
> > > Thank you very much for your kind reminder, we have uploaded the revised version now. We hope our answer can solve your question, thank you again.

---

> > > > ### Comment · Reviewer_oJee · 2024-11-26
> > > > **+1 point**
> > > >
> > > > Looks good to me. I moved my score up to 6.
> > > >
> > > > Best

---

> > > > > ### Author Response · Authors · 2024-11-27
> > > > > **Thanks for your support**
> > > > >
> > > > > Dear Reviewer oJee,
> > > > >
> > > > > Many thanks for raising the score! Your constructive feedback has been invaluable in helping us refine our work. We sincerely appreciate your recognition of the significant contributions of InterDance.
> > > > >
> > > > > Your acknowledgment holds great importance to us,  thank you again for your time and support!
> > > > >
> > > > > Warm Regards,
> > > > >
> > > > > The Authors

---

> ### Author Response · Authors · 2024-12-01
> **Follow up with reviewer oJee**
>
> Dear Reviewer oJee,
>
> We sincerely appreciate the time and effort you have dedicated to reviewing our paper. Your insightful questions and valuable feedback have been extremely beneficial.
>
> During this period, we have made consistent efforts to add supplementary experiments and videos to enhance the quality of our work. To better response your suggestion regarding the *Baseline of Weaknesses*, we trained and tested our method on the DD100 dataset to provide a more fair apple-to-apple comparison with Duolando. The results are as follows, on the DD100 benchmark, our method also outperforms Duolando on most metrics. Especiallyin the $\text{DIV}_g$, $\text{CF}$ and $\text{BED}$, our method significantly outperforms than Duolando.
>
> | Method | $\text{FID}_k \downarrow$ | $\text{FID}_g  \downarrow$ | $\text{DIV}_k  \uparrow $ | $\text{DIV}_g \uparrow$ | $\text{FID}_{cd} \downarrow$ | $\text{DIV}_{cd} \uparrow$ | $\text{CF}\uparrow$| $\text{BED} \uparrow$ | $\text{BAS}\uparrow$ |
> | --- | --- | --- | --- | --- | --- | --- | --- | --- | --- |
> | Duolando| 43.33| **35.30** | **10.24** | 7.04 | 5.34 | 11.10 | 57.67\%  | 0.2962 | 0.2116 |
> | Ours       | **41.67**| 46.27 | 9.93 | **12.75** | **5.19** | **12.41**   |**62.24\%**   | **0.4208** | **0.2136** |
>
> We hope that these additional experiments have further addressed your concerns. If you have any other questions or suggestions, please do not hesitate to let us know.
>
> Thank you once again for your invaluable contribution to our research.
>
> Warm Regards,
>
> The Authors

---

### Author Response · Authors · 2024-12-03
**Gloabl Response**

Dear Reviewers and Area Chairs,

We thank all the reviewers for the thorough reviews and valuable feedback. We are pleased to hear that the dataset is recognized as a valuable contribution to the community (all reviewers); that our motion representation correctly predicts contact (Reviewer oJee), is designed to better capture interactions (Reviewer yXpV), is essential for modeling interactive motions (Reviewer UHsx), makes sense (Reviewer 5vWh), and help on close close interactions (Reviewer jdiH). Regarding our method, four of the reviewers (oJee, yXpV, UHsx, jdiH) have positively commented our method on the strength section.  We also appreciate the recognition of our comprehensive quantitative evaluations (Reviewers UHsx, 5vWh, and jdiH) and that the paper is well-written and easy to follow (Reviewers yXpV, 5vWh, and jdiH).

Your recognition and encouraging words are of great significance to our work! During the rebuttal period, we made the following improvements based on the reviewers' comments:

1. Added more comprehensive baselines, conducted ablation studies, and compared our method with others on the DD100 benchmark. These experiments demonstrated that our method outperforms previous approaches.

2. Added additional qualitative video results, showcasing not only reactive and duet dance generation but also the effects of long dance generation.

3. Addressed reviewers' questions and made corresponding revisions in the paper, with the modified text highlighted in blue.

4. Provided more details about the dataset.

Your suggestions have been invaluable in enhancing the quality of this work. Finally, as the rebuttal ddl is coming, we would greatly appreciate it if you could review our response further.

Thanks again you for your time and support!

Warm Regards,

The Authors

---

### Meta-Review · Area_Chair_Dd4Z · 2024-12-20

**Metareview:**

The submission is about 3D dance generation.  None of the five reviewers were enthusiastic about the submission or would like to champion its acceptance, probably due to concerns about the limited technical innovations and the lack of qualitative results.  While the authors made significant attempts during rebuttal, reviewers remained unexcited.  The AC read the submission, reviews, and rebuttals, and agreed with the reviewers that the submission's contributions are insufficient for ICLR.

**Additional Comments On Reviewer Discussion:**

Reviewers all agreed that they would not like to champion this submission.

---

### Decision · Program_Chairs · 2025-01-22

Reject